# Architecture and regulation of filamentous human cystathionine beta-synthase

Thomas J. McCorvie [1,5] ✉, Douglas Adamoski [2], Raquel A. C. Machado [2], Jiazhi Tang[3], Henry J. Bailey[1,6], Douglas S. M. Ferreira[1,3], Claire Strain-Damerell [1,7], Arnaud Baslé[3], Andre L. B. Ambrosio [4], Sandra M. G. Dias [2] & Wyatt W. Yue [1,5] ✉

Cystathionine beta-synthase (CBS) is an essential metabolic enzyme across all domains of life for the production of glutathione, cysteine, and hydrogen sulfide. Appended to the conserved catalytic domain of human CBS is a regulatory domain that modulates activity by S-adenosyl-L-methionine (SAM) and promotes oligomerisation. Here we show using cryo-electron microscopy that full-length human CBS in the basal and SAM-bound activated states polymerises as filaments mediated by a conserved regulatory domain loop. In the basal state, CBS regulatory domains sterically block the catalytic domain active site, resulting in a low-activity filament with three CBS dimers per turn. This steric block is removed when in the activated state, one SAM molecule binds to the regulatory domain, forming a high-activity filament with two CBS dimers per turn. These large conformational changes result in a central filament of SAM-stabilised regulatory domains at the core, decorated with highly flexible catalytic domains. Polymerisation stabilises CBS and reduces thermal denaturation. In PC-3 cells, we observed nutrient-responsive CBS filamentation that disassembles when methionine is depleted and reversed in the presence of SAM. Together our findings extend our understanding of CBS enzyme regulation, and open new avenues for investigating the pathogenic mechanism and therapeutic opportunities for CBS-associated disorders.

The pyridoxal 5′-phosphate (PLP)-dependent enzyme CBS catalyses the condensation of serine and homocysteine to form cystathionine[1]. Playing a pivotal role in the transsulfuration pathway and redox regulation[2,3] and being linked to one-carbon metabolism, CBS produces cysteine, glutathione, as well as the gaseous transmitter hydrogen sulfide[4,5] ($H_2S$). CBS has recently gained interest as a therapeutic target as inhibition of $H_2S$ production has been suggested for the treatment of specific cancers[6] and Down's Syndrome[7]. Inherited loss-of-function mutations of CBS result in classical homocystinuria (HCU), the most common inborn error of sulfur metabolism[8]. Classical homocystinuria has been recognised as a protein misfolding disorder as many of the known pathogenic mutations result in aggregation and degradation of the CBS enzyme[9,10].

Human CBS adopts a unique multi-domain structure: an N-terminal haem binding domain (residues 38–74), a central PLP-dependent catalytic domain (CD, residues 75–382), and a C-terminal

[1]Nuffield Department of Clinical Medicine, Centre for Medicines Discovery, University of Oxford, Oxford OX3 7DQ, UK. [2]Brazilian Biosciences National Laboratory, Brazilian Center for Research in Energy and Materials, 13083-970 Campinas, Brazil. [3]Biosciences Institute, The Medical School, Newcastle University, Newcastle upon Tyne NE2 4HH, UK. [4]Sao Carlos Institute of Physics, University of Sao Paulo, Sao Carlos, SP, Brazil. [5]Present address: Biosciences Institute, The Medical School, Newcastle University, Newcastle upon Tyne NE2 4HH, UK. [6]Present address: Faculty of Medicine, Institute of Biochemistry II, Goethe University Frankfurt, Frankfurt, Germany. [7]Present address: Research Complex at Harwell, Harwell Science and Innovation Campus, Didcot OX11 0FA, UK. ✉e-mail: thomas.mccorvie@newcastle.ac.uk; wyatt.yue@newcastle.ac.uk

regulatory domain (RD, residues 411–551)[5,11,12] (Fig. 1a). The RD adopts the evolutionarily conserved Bateman module, consisting of two tandem *CBS* motifs (CBS-1 and CBS-2) and a surface-exposed loop (residues 516–525) extending from a CBS-2 motif (Supplementary Fig. 1a). Bateman modules act as a regulatory sensor in response to the binding of predominantly adenosine-containing ligands[13] and are present in enzymes across the three domains of life[14]. In mammals, the activity of CBS is increased by the binding of SAM to the Bateman module[15]. The enzyme is therefore proposed to transition between the basal state in the absence of SAM and the SAM-bound activated state[15].

Crystal structures of human CBS[8–10] have been determined of CD alone (CBS^CD), of full-length protein engineered with the RD loop deletion (CBS^Δ516–525)[16–18], and of RD alone with the loop deletion (CBS^RDΔ516–525) complexed with SAM[17]. Together, they depicted human CBS as a homodimer, where both CD and RD can dimerise independently. The structural data also revealed the molecular mechanism of SAM activation, whereby the Bateman module in the RD acts as an autoinhibitory cap to the CD, and upon binding SAM, undergoes a conformational change relieving the inhibition to allow active site access for catalysis (Supplementary Fig. 1)[17,18]. However, these structures do not consider various reports that the RD promotes the formation of CBS tetramers and higher-order oligomers with a high tendency to aggregate[19–22]. Indeed, CBS oligomerisation and SAM response vary across the animal kingdom[12]; it is unclear how human CBS oligomerises and their relation to SAM allosteric regulation are unknown[13,14,23].

Additionally, mutations on or deletion of the Bateman module can rescue the most common homocystinuria-associated mutations, but the structural information so far has given limited insight[24–26]. Efforts have also been made to develop new small molecule therapies targeting CBS[7,26,27], but the lack of full-length enzyme structure has been a possible hindrance. In this work, we used a combination of biophysical techniques and cryo-electron microscopy (Cryo-EM), showing that human CBS full-length protein polymerises as a filament adopting two very different morphologies dependent on SAM binding and that polymerisation plays a role in both enzyme activation and stability. Furthermore, we confirm CBS forms oligomers in cells using fluorescence microscopy.

## Results
### Cryo-EM of full-length human CBS reveals a filamentous architecture

We pursued the cryo-EM structure of full-length CBS initially using two constructs: one where an N-terminal His-tag was removed during purification (CBS^FL) and one with a permanent C-terminal His-tag (CBS^FL-CHis) (Supplementary Fig. 2a). In analytical gel filtration on a Superose 6 column, both CBS^FL and CBS^FL-CHis proteins eluted as a broad peak, suggestive of different oligomeric states with molecular weights larger than tetramers possibly up to the range of 18–37 mDa and potentially even larger (Supplementary Fig. 2b, e). Similar observations of larger-than-tetrameric CBS^FL oligomers and even higher

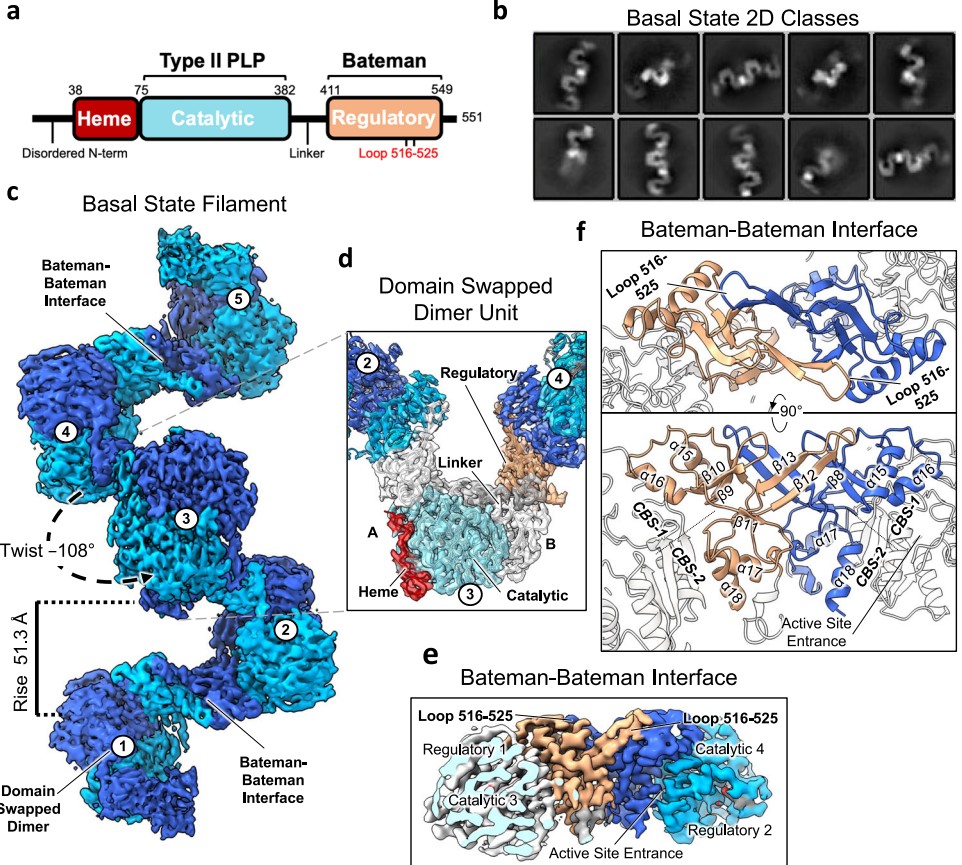

**Fig. 1 | Cryo-EM structure of the basal state of CBS. a** Domain diagram of human CBS. **b** Initial representative 2D classes of CBS in the basal state. **c** Cryo-EM helical reconstruction of the CBS filament in the basal state at a resolution of 3.9 Å. Individual domain-swapped dimers (coloured in two shades of blue for the two monomers) are numbered. **d** Single particle reconstruction focused on one CBS dimer unit at a resolution of 3.0 Å. Dimers 2 and 4 are coloured as in (**c**). For dimer 3, monomer A is coloured according to domain organisation as shown in (**a**), and monomer B is coloured white. **e** Close-up view of the EM density at the filament interface. This is formed by regulatory domain interactions from neighbouring domain-swapped CBS dimers. The active site entrance of one catalytic domain is indicated. **f** Model of the Bateman–Bateman interface formed by regulatory domain interactions. The C2 symmetry is apparent and shows the key role of the loop 516–525. The active site entrance of one catalytic domain is indicated.

molecular weight oligomers for CBS$^{FL-CHis}$, were made in Coomassie-stained clear-native and blue-native PAGE experiments (Supplementary Fig. 2c, d). As control, CBS$^{\Delta516-525}$, CBS$^{CD}$, and CBS$^{RD\Delta516-525}$ behaved as expected dimers[11,16,17]. Micrographs of full-length CBS from both constructs in vitreous ice clearly showed the presence of flexible filaments (Supplementary Figs. 3–6) of varying lengths agreeing with 2D classes (Fig. 1b, Supplementary Figs. 3c, 4b, 5b, 6b). Multiple maps were processed using both helical (Supplementary Figs. 3, 5, 7) and single particle (Supplementary Figs. 4, 6, 7) reconstruction to resolutions of between 3.9 and 3.0 Å. Both constructs resulted in maps with virtually identical conformations (Supplementary Fig. 8a), and both were used for modelling.

The CBS filament adopts a left-handed helical architecture with a twist of −108° and a rise of 51 Å (Fig. 1c, Supplementary Figs. 3, 5). The domain-swapped dimer, previously observed in crystal structures, is the repeating unit, and helical formation is driven by inter-dimeric RD interactions, i.e., between the RDs of neighbouring subunits of the filament. Here the RD sits atop the active site entrance of the catalytic domain hindering substrate access (Fig. 1e, f). Loop 516–525, a β-turn-β extension from the CBS-2 motif in the conserved Bateman module, plays a key role in driving oligomerisation. Specifically, one RD each from two neighbouring CBS dimers interact in a butterfly-like dimeric arrangement, related by C2 symmetry, such that the loop 516–525 from an RD of one dimer forms a clasp around an RD of the neighbouring dimer. This packing arrangement, burying a total surface area of ~1550 Å$^2$ (Fig. 1e, f, Supplementary Fig. 8b), is stabilised through two interfaces composed entirely of the RDs. The first interface is between the CBS-2 and CBS-1 motifs, involving loop 516–525 of one dimeric subunit and α-helix 15 of the neighbouring dimeric subunit. Here, main-chain interactions are mediated between β-strand 12 residues 516–519 of one RD and β-strand 8 residues 422–426 of the adjacent RD (Fig. 1f, Supplementary Fig. 8d). This arrangement results in residue Tyr518 from the oligomerisation loop slotting into a hydrophobic pocket formed by Leu423, Val425, Ile429, and Ile437 of α-helix 15 (Supplementary Fig. 8d). The second interface involves inter-dimeric RD contacts, between the two neighbouring CBS-2 motifs. Here RD residues Leu419, Leu492, Met529, and Phe531 of one dimeric subunit form hydrophobic packing interactions with the equivalent residues of the neighbouring dimeric subunit (Supplementary Fig. 8e). Due to these interactions, both Ala421 and Pro422 are shifted from their positions as observed in the previous crystal structures of CBS$^{\Delta516-525}$ (Supplementary Fig. 8f)[16–18]. Altogether, the two sites of RD interactions demonstrate how loop 516–525 is the main driver of CBS oligomerisation.

## CBS degrades into tetramers and dimers

Though both full-length CBS constructs resulted in highly similar filament maps, one key difference was the observed protein degradation and the presence of slightly shorter oligomers of the CBS$^{FL}$ construct without a permanent His-tag (Supplementary Fig. 2a, b). Consequently, along with the filament classes, we observed 2D averages of this construct that represented these degradation products (Supplementary Fig. 6a, b). One collection of classes appeared to be the catalytic domain dimer alone, suggesting that the RD was degraded from some full-length protein, although we could not obtain a reasonable reconstruction due to its small size (~80 kDa) (Supplementary Fig. 9a). Another collection of classes resulted in a 3.8 Å map of a degraded CBS tetramer (Supplementary Figs. 6, 7, 9). Here two degraded heterodimers, composed of one full-length protomer and one RD-degraded catalytic domain protomer (Supplementary Fig. 9b), interact through a single Bateman–Bateman interface, highly like the arrangement seen in the intact filament. The degradation of the RD renders the active site loops within the catalytic domain of the full-length CBS in a more opened state (Supplementary Fig. 9c). It is unknown if this degradation is due to

recombinant expression or is related to a biological function of CBS. Further studies are needed to determine its significance.

## SAM binding transforms the morphology of the CBS filament

The global methyl donor SAM functions as an allosteric activator of human CBS activity[15]. In our biophysical characterisation assays (Supplementary Fig. 2c, d), SAM had no apparent effect on the oligomeric status of all CBS constructs, agreeing with previous reports[23]. Since oligomerisation occurs without SAM, we hypothesised that SAM could modulate the morphology of the CBS filament as part of its role as an allosteric activator. To this end, cryo-EM was used to analyse CBS$^{FL-CHis}$ in the presence of SAM. Micrographs showed that CBS retains a filamentous architecture in the presence of SAM but with a significantly altered morphology, as shown by 2D classes (Fig. 2a, Supplementary Fig. 10a, c). Obtaining a 3D reconstruction took considerable effort due to the highly flexible nature of the filaments (Supplementary Figs. 10, 11, 12). Through helical refinement, we generated a global map at 4.0 Å resolution which reveals a central helical stalk decorated with highly flexible lobes (Fig. 2b). To aid in model building, we applied masks and performed local refinement that resulted in local maps for the central stalk at 4.1 Å resolution and for the flexible domain at 8.3 Å resolution (Supplementary Fig. 11).

These maps allowed us to model the entire filament by docking one catalytic domain dimer into each flexible lobe (PDB: 4PCU) and repeating units of the regulatory domain dimer (PDB: 4UUU) into the central stalk. The overall morphology of the resulting filament is drastically different in comparison to the basal state, as reflected by the 66% increase in twist (−178.6°) and 10% decrease in rise (46.9 Å) of the filament in the presence of SAM (Fig. 2b, Supplementary Fig. 13a). Previous crystal structures of the non-filamentous CBS$^{\Delta516-525}$ dimers showed that SAM binding to the RD elicits dis-association from the CD and subsequently its homo-dimerisation, thereby freeing the active site access for catalysis (Supplementary Fig. 1a)[17,18]. In the context of the full-length enzyme elucidated here, the SAM-mediated conformational change creates a central filament stalk composed of repeating units of the SAM-bound RD dimer, arranged in an antiparallel "daisy-chain" like fashion due to the interactions of the loop 516–525 with the neighbouring subunit (Fig. 2b, c). The central filament stalk is decorated by highly flexible CDs, where the active site entrance loops are free to open and hence increase the accessibility for substrates (Supplementary Fig. 13b)[18].

Comparing our basal and activated filaments, the SAM-mediated conformational change to the RD is highly agreeable with that observed in the crystal structures of CBS$^{\Delta516-525}$ dimers where there is a relative 18° rotation between the CBS-1 and CBS-2 motifs caused by SAM binding (Fig. 2d)[17,18]. Observing this interface of CBS-1 and CBS-2 in isolation, the conformational change is mainly localised in the CBS-1 motif and appears as an unfurling of the "butterfly wings" of this dimeric arrangement (Supplementary Movie 1). This results in α-helix 15 being displaced by 8 Å and α-helix 16 by as much as 11 Å from their original position (Fig. 2d). Interestingly, loop 516–525 from the neighbouring subunit also moves 3.0 Å to maintain its interactions with residues 422–426 and α-helix 15 (Fig. 2d). Additionally, alignment to the central helical Z-axis and a simple morph of the global basal and activated models show that the large conformational change is possible in the filament with fewer clashes when the CD moves in concert with the RDs (Supplementary Fig. 13d, Supplementary Movie 2).

## Only one SAM binding site is observed in CBS filaments

Each Bateman module, assembled from the tandem CBS-1 and CBS-2 motifs, contains, in principle, two ligand-binding sites (S1 and S2) related by dyad symmetry (Supplementary Fig. 1a)[16]. In our activated filament maps, ligand density was present for SAM only at the S2 site (Figs. 2d, 3a). No apparent and interpretable density was found for SAM at the S1 site nor at the filament interfaces. This 1:1 (one SAM to

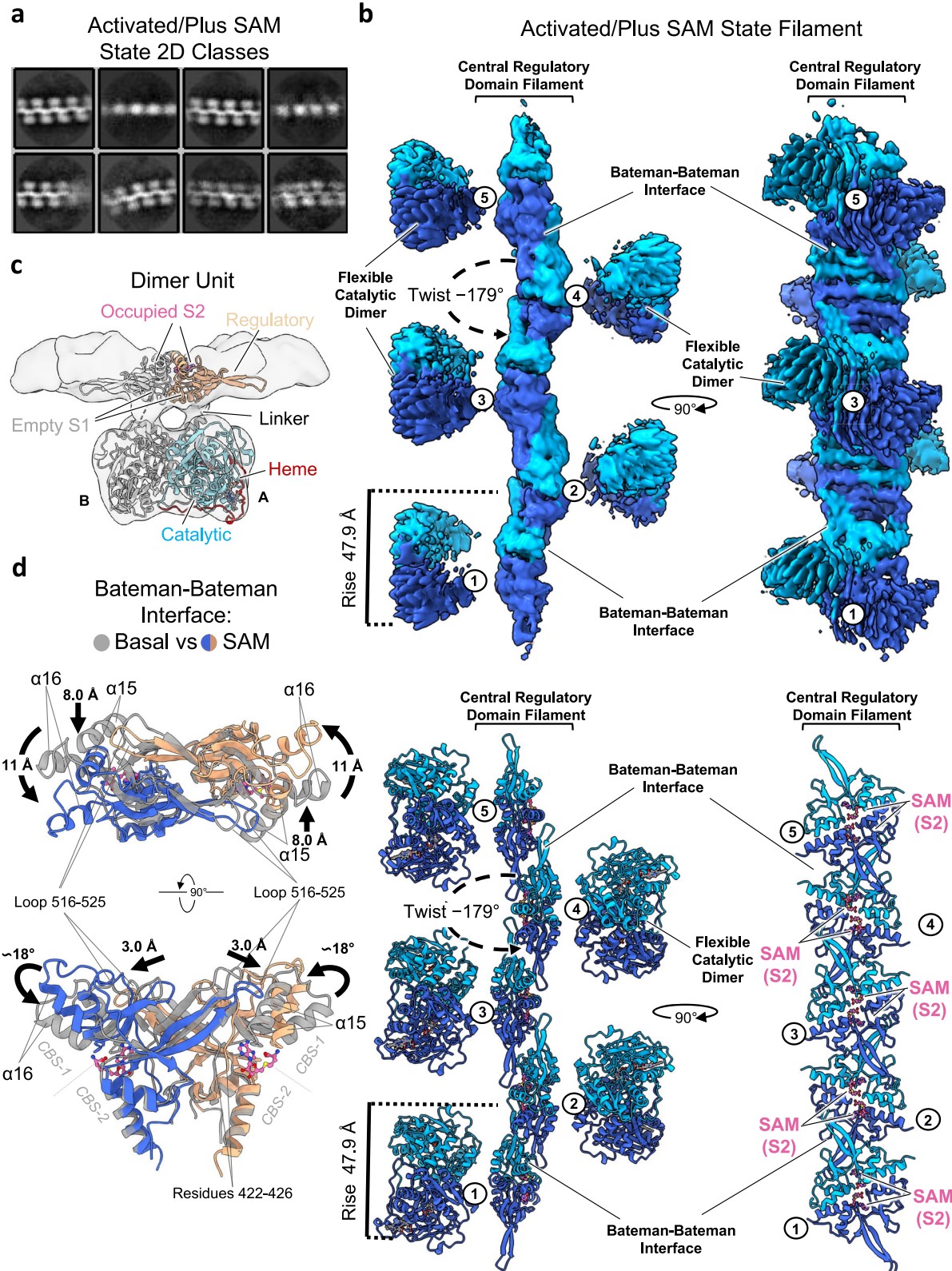

**a** Activated/Plus SAM State 2D Classes

**b** Activated/Plus SAM State Filament

**c** Dimer Unit

Occupied S2 · Regulatory · Empty S1 · Linker · Heme · Catalytic · B · A

**d** Bateman-Bateman Interface: ● Basal vs ◐ SAM

Central Regulatory Domain Filament

Bateman-Bateman Interface

Flexible Catalytic Dimer

Twist −179°

Rise 47.9 Å

SAM (S2)

α16 · α15 · 8.0 Å · 11 Å · Loop 516-525 · 3.0 Å · ~18° · CBS-1 · CBS-2 · Residues 422-426

**Fig. 2 | Cryo-EM structure of the SAM-bound activated state CBS. a** Initial representative 2D classes of the activated CBS filament in the presence of SAM. **b** Global helical cryo-EM map of the activated SAM-bound CBS$^{FL}$ filament at a resolution of 4.1 Å and model. The catalytic domains are omitted for clarity in the bottom right panel. Individual dimers are numbered. **c** Single particle reconstruction focused on one CBS dimer unit at a resolution of 8.0 Å. One monomer is coloured as in Fig. 1a. **d** Structural alignment of the Bateman-Bateman interface in the basal and activated states showing the conformational change due to SAM binding.

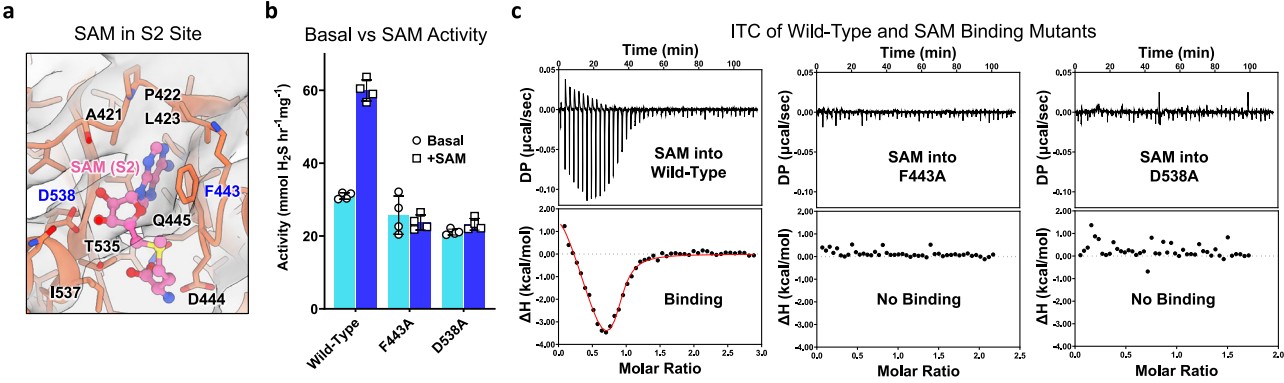

**Fig. 3 | Mutation of S2 site residues reduces SAM activation and binding. a** Cryo-EM density of SAM-bound in the S2 site. One Bateman module (salmon) bound with one SAM (pink) in the S2 site is shown. Interacting residues are represented as sticks and SAM is represented as balls and sticks. **b** $H_2S$-producing activity of wild-type, F443A, and D538A CBS$^{FL-CHis}$ as purified (basal) and in the presence of 300 μM SAM (+SAM). Mean values and error bars are ±s.d. of $n = 4$ technical repeats. **c** ITC titrations of SAM against wild-type, F443A, and D538A CBS$^{FL-CHis}$. Each titration is representative of two independent experiments. Red line represents the fit of experimental data (dots) into the two-site binding model. Source data for **b** and **c** are provided as a Source Data File.

one CBS protomer) stoichiometry is identical to previously observed CBS$^{\Delta516-525}$ crystal structures where only the S2 site was occupied by SAM[17,18]. These structures suggest that Phe443 and Asp538 at the S2 site are key residues in SAM binding (Fig. 3a, Supplementary Fig. 12b)[17]. Therefore, to confirm the 1:1 binding of SAM, we purified the mutants F443A and D538A and initially tested their enzymatic response to SAM. These mutants showed a low basal activity, but their activity could not be stimulated by SAM-like wild-type (Fig. 3b).

We next performed isothermal titration calorimetry (ITC) of wild-type CBS$^{FL-CHis}$ against SAM which demonstrated two apparent binding events at ~160 and ~600 nM, in agreement with previous reports[23,28,29]. In contrast, the S2 site substitutions F443A and D538A are sufficient to eliminate both SAM binding events (Fig. 3c, Supplementary Fig. 14b). This suggests that no other SAM sites exist in the filament beyond S2, but it is also possible that mutation of these residues could alter another putative cryptic SAM binding site indirectly. We therefore reasoned that the two apparent binding events could be attributed to a combination of the binding of one SAM to the S2 site and the resulting conformational rearrangement into the activated state. In support of this interpretation, the ITC of CBS$^{FL}$, where the protein can respond to both ligand binding and conformational changes, demonstrated two apparent binding events (Supplementary Fig. 15a, b). CBS$^{\Delta516-525}$, where the protein can respond to both ligand binding and conformational changes, also presented two events, though this construct is dimeric and will likely not form a putative second SAM site at a higher order oligomeric interface (Supplementary Figs. 2, 15a, b). In contrast, the RD alone protein CBS$^{RD\Delta516-525}$, which only responds to ligand binding and does not form higher-order oligomers, presented only one event at ~4 μM (Supplementary Figs. 2, 15a, b). Oddly the first binding event of CBS$^{FL-CHis}$ is endothermic (positive enthalpy) and increases order (negative entropy), in contrast to the constructs without a C-terminal His-tag, which have a first binding event that is more exothermic (negative enthalpy) and increases disorder (positive entropy) (Supplementary Figs. 14b and 15). Though we do not know the exact reason for this phenomenon, this difference is likely due to the longer oligomers and stabilising effect of the C-terminal His-tag (Supplementary Figs. 2 and 14a)[30]. Overall, our data suggests that CBS likely binds SAM in a 1:1 manner and that our observed filament structures only reveal S2 as a SAM binding site.

## Filament formation effect on cooperativity of SAM activation and CBS stability

Polymerisation of metabolic enzymes, such as involving filament formation, has been linked to their regulation and stabilisation[31,32], and we hypothesised that the assembly of human CBS into a filament could

play a similar role that facilitates enzyme catalysis. Our observation of loop 516−525 moving in tandem with the conformational change of α-helix 15 (Fig. 2d) in the presence of SAM suggests potential crosstalk communication between regulatory domains from neighbouring CBS proteins (inter-dimer) in the filament (Supplementary Movie 1). To investigate potential cooperativity within the CBS filament, we char-acterised CBS activity by measuring $H_2S$ production from the condensation of cysteine and homocysteine. $K_m$ values for both homocysteine and cysteine were essentially identical for the four constructs at ~0.3 and ~20 mM, respectively. CBS$^{FL}$, CBS$^{FL-CHis}$, and CBS$^{\Delta516-525}$ were allosterically activated by SAM, whereas CBS$^{CD}$ was not (Supplementary Fig. 16a). By titrating SAM, we found that the responsive constructs showed a ~2-fold increase in activity, exhibiting $K_{act}$ for SAM of ~26.0−36.0 μM (Fig. 4a, Supplementary Fig. 16b) agreeing with reported values[11,25,33]. The activation of CBS$^{FL}$ and CBS$^{FL-CHis}$ exhibits a Hill coefficient ($n_{Hill}$) of 3.0−3.6, whereas the non-filamentous CBS$^{\Delta516-525}$ presented a lower $n_{Hill}$ of 2.0 (Fig. 4a, Supplementary Fig. 16b). Comparison of the $n_{Hill}$ values gave $p$-values of 0.2218 (CBS$^{FL}$ vs. CBS$^{\Delta516-525}$) and 0.3382 (CBS$^{FL-CHis}$ vs. CBS$^{\Delta516-525}$) in Student's $T$-test, suggesting that further studies are likely needed to determine their significance.

Next, we determined if filament formation alters CBS stability by using thermal shift. The CBS$^{\Delta516-525}$ protein, which does not form fila-ments, is less thermostable than CBS$^{FL}$ by ~5 °C, confirming that fila-mentation increases stability. We also found that CBS$^{FL-CHis}$ is more thermostable than CBS$^{FL}$ by ~3 °C (Fig. 4b, Supplementary Fig. 14a), possibly because it is forming longer oligomers and is less degraded than CBS$^{FL}$ (Supplementary Fig. 2)[30]. Moreover, it is known that the activity of human CBS can be increased by thermal activation, likely due to the denaturation of the regulatory domain that relieves its autoinhibitory effect[15,19,28]. Repeating this assay on both CBS$^{FL}$ and CBS$^{\Delta516-525}$, we determined $T_m$ values of 49.5 and 43.7 °C respectively, showing that CBS$^{\Delta516-525}$ is more prone to thermal activation and that its regulatory domain is less stable (Fig. 4c). These findings suggest that polymerisation stabilises the regulatory domain probably to maintain CBS in the basal state conformation.

## CBS filament formation in cells

To investigate CBS filamentation in situ, we employed fluorescence microscopy across various cell lines, including non-transformed (MEF, hFB), prostate cancer (PC-3), and breast cancer (BT549, MCF7, MDA-MB-231) cells (Fig. 5a). Cells labelled with full-length CBS (mKO2-CBS$^{FL}$) exhibited an aggregated fluorescence pattern consistent with cryo-EM-identified filament structures. In PC-3 cells, this pattern was notably responsive to culture conditions (Fig. 5b, c). SAM supplementation in

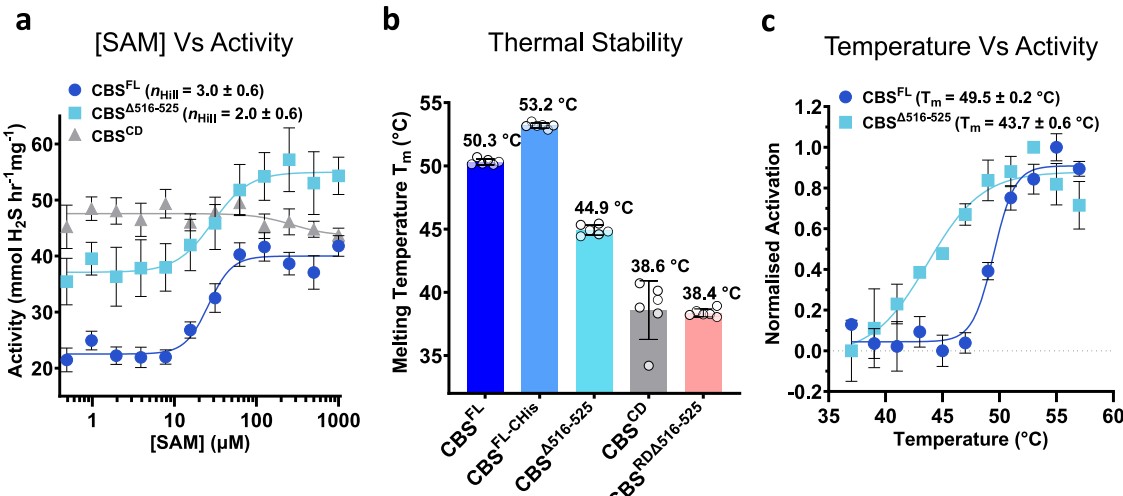

**Fig. 4 | Effect of filament formation on SAM cooperativity and stability of CBS.** **a** H$_2$S-producing activity of CBS$^{FL}$, CBS$^{\Delta 516-525}$, and CBS$^{CD}$ in response to increasing amounts of SAM. Mean values and error bars are ±s.d. of $n$ = at least 4 technical repeats. **b** Thermal stability of CBS$^{FL}$, CBS$^{FL-CHis}$, CBS$^{\Delta 516-525}$, CBS$^{CD}$, and CBS$^{RD\Delta 516-525}$. Mean values and error bars are ±s.d. of $n$ = 6 technical repeats. **c** Thermal activation of CBS$^{FL}$ and CBS$^{\Delta 516-525}$ activity. Mean values and error bars are ±s.d. of $n$ = at least 4 technical repeats. Source data for **a**–**c** are provided as a Source Data File.

complete media did not significantly alter the fluorescence pattern ($p > 0.05$). However, removing glutamine, cystine, and methionine markedly reduced the percentage of cells with filamentous fluorescence (from 64.7 ± 2.3% to 25.6 ± 2.8%, $p < 0.0001$), which was reversed by SAM addition (increased to 55.9 ± 4.3%, $p < 0.0001$). Methionine depletion alone sufficiently reduced the filamentous pattern (to 21 ± 2.7%, $p < 0.0001$), with SAM addition restoring it (to 69.2 ± 2.4%, $p < 0.0001$). mKO2-CBS variants F443A and D538A, with impaired SAM binding and activation, showed less disaggregation following nutrient deprivation (Supplementary Fig. 17). Compared to mKO2-CBS$^{FL}$ (Fig. 5d), mKO2-CBS$^{\Delta 516-525}$ disrupted filament formation, forming smaller periphery puncta (Fig. 5e). These findings underscore CBS filamentation's critical role in cellular responses to nutrient changes.

## Discussion

Conflicting reports of human CBS oligomerisation, evidenced by a variety of biophysical and immunoblotting techniques using both recombinant and endogenous sources of the protein, have plagued the literature since its initial characterisation[10,11,20–23,33,34]. Here we have shown that human CBS oligomerises into filaments, adopting (at least) two distinct architectures, respectively, for the basal and activated states. Chiefly our findings reflect the initial reports of CBS purified from human liver, where it was shown to form large oligomers with a tendency to both aggregate and degrade to a more active state[22]. This discovery of filamentation evaded past crystallographic studies involving an engineered protein that removes a surface loop (residues 516–525)[16–18] now revealed to be key to polymerisation, alongside the assumption that CBS is predominantly a tetrameric protein[12]. Our observation of CBS filaments with heterogeneous lengths in cryo-EM micrographs, therefore, sufficiently explains previous findings of a mixture of CBS oligomers. Due to this heterogeneity, the reported tetrameric state of CBS is likely to be two dimers interacting through a single Bateman module pair but additionally could also be formed via the degradation of the Bateman module that we have shown here (Supplementary Fig. 9).

The link between the oligomeric state and SAM activation has also had conflicting reports. Originally shown to bind only one SAM molecule per monomer, more recent ITC studies suggested a two-site model where a kinetically stabilising high-affinity SAM binding site would be formed from oligomerisation, while enzyme activation is driven by SAM binding to the lower affinity S2 site observed in dimeric CBS$^{\Delta 516-525}$

crystal structures[23,28,35]. Our cryo-EM structure of the SAM-bound activated state (Fig. 2), mutagenesis data (Fig. 3), and previous biophysical analyses[17] all tentatively suggest that only the S2 site exists to bind SAM. Though our ITC analysis of full-length CBS fits the two-site model (Fig. 3c, Supplementary Figs. 14, 15), we reason that the thermograph reflects not only SAM binding but also the structural rearrangements that have to occur for activation. This is especially so, as we observe that dimeric CBS$^{\Delta 516-525}$ also fits the two-site model even though this construct cannot form higher oligomers (Supplementary Figs. 2 and 15) and its crystal structure only shows one SAM molecule bound per monomer (Supplementary Fig. 1a). This transition from basal to activated states requires a significant conformational change and likely follows a multistep process (Supplementary Fig. 13d, Supplementary Movie 2). Conformational changes due to ligand binding are known to be a major contributor to heat capacity changes[36], and there is precedence for ligand binding to Bateman modules to diverge from a simple one-site binding model when dimerisation of Bateman modules occurs[37]. Therefore, we regard our ITC data as relative measurements of both SAM binding and conformational changes. We acknowledge a possibility that the elusive 'high-affinity kinetically stabilising site' could be located at the flexible interface between the regulatory domains and hence difficult to be resolved in cryo-EM studies. However, the ITC of dimeric CBS$^{\Delta 516-525}$ (Supplementary Fig. 15) and our mutagenesis data of S2 site residues do not appear to support it (Fig. 3b). It is also possible that mutation of S2 site residues could alter indirectly a putative cryptic binding site at the oligomeric interface especially as these mutants affect oligomerisation in our cell-based assays (Supplementary Fig. 17). As such further studies are clearly required to clarify the presence or absence of this putative cryptic SAM binding site. While this putative kinetically stabilising site has been suggested for drug discovery[28,35], our findings nonetheless suggest alternative frameworks in the context of filament formation (instead of the previously suggested tetrameric state) that should be considered for targeting the CBS regulatory domain (discussed below).

This study now firmly places human CBS in the growing membership of filamentous metabolic enzymes (Fig. 6a). Higher order oligomerisation in response to signal transduction from ligand (nutrient) binding or stress has been shown for many eukaryotic metabolic enzymes such as acetyl-CoA carboxylase (ACC), inosine-5′-monophosphate dehydrogenase (IMPDH), and cytidine triphosphate synthase (CTPS)[31,32]. The yeast CBS orthologue, *S. cerevisiae* Cys4p, forms

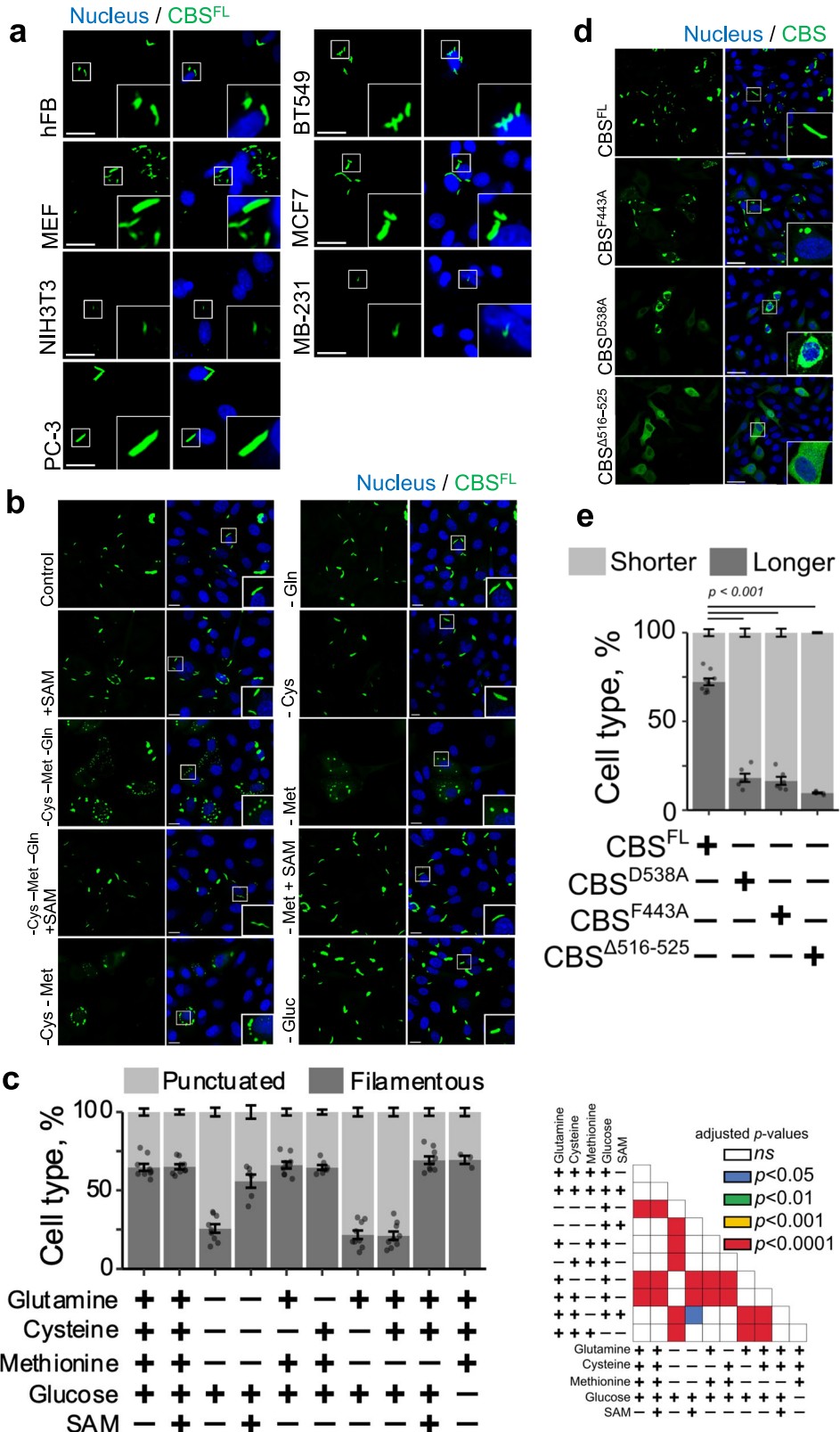

punctate foci during stationary phase in response to nutrient levels[38], an observation that suggests filamentation[32]. However, yeast Cys4p, contrary to human CBS, does not undergo allosteric feedback activation in response to SAM[12] and does not contain the same oligomerisation loop in the regulatory domain (Supplementary Fig. 18). Therefore, it remains unclear if SAM constitutes the signal or driver for Cys4p oligomerisation.

For human CBS, filament formation for the recombinant protein occurs in the absence and presence of SAM. Additionally, in human cells, we observed fluorescence of transfected CBS in an aggregated profile that is consistent with higher-order oligomerisation and completely dependent upon the integrity of the RD loop 516–525. Importantly, this aggregation pattern in the cell also responds to metabolite manipulation, as fluorescence becomes diffuse when methionine is

**Fig. 5 | Cellular dynamics of CBS filamentation and methionine dependence.**
**a** Filament-like structures of mKO2-CBS[FL] in various cell lines, including mouse embryonic fibroblasts (MEF), human fibroblasts (hFB), prostate cancer (PC-3), and breast cancer cells (BT549, MCF7, MDA-MB-231). Scale bar: 30 µm; inset: ×3 magnification. At least 20 cells per lineage were evaluated. **b** PC-3 cells transfected with mKO2-CBS[FL], incubated in complete or nutrient-depleted medium for 8 h. Scale bar: 20 µm; inset: ×2 magnification. **c** Fraction of cells with predominant filamentous or punctuated morphology based on fractal-D complexity. The range of evaluated cells per condition is 163–2664, totalling 166,562 cells. Statistical

significance is shown via one-way ANOVA and Tukey's test. Significance for (**c**) on the right. **d** mKO2-CBS[FL] and its variants transfected in PC-3 cells; morphology evaluated after 24 h. Scale bar: 50 µm; inset: ×3 magnification. **e** Quantification of cellular response in (**d**), based on the majority of shorter or longer cytoplasmic objects, measured by perimeter. The range of evaluated cells per condition is 126–2474, totalling 49,248 cells. One-way ANOVA p-value is <0.00001, and all labelled Tukey HSD comparisons had p-values < 0.00001. Mean and s.e.m. are shown for all plots. NS not significant. Source data for **c** and **e** are provided as a Source Data File.

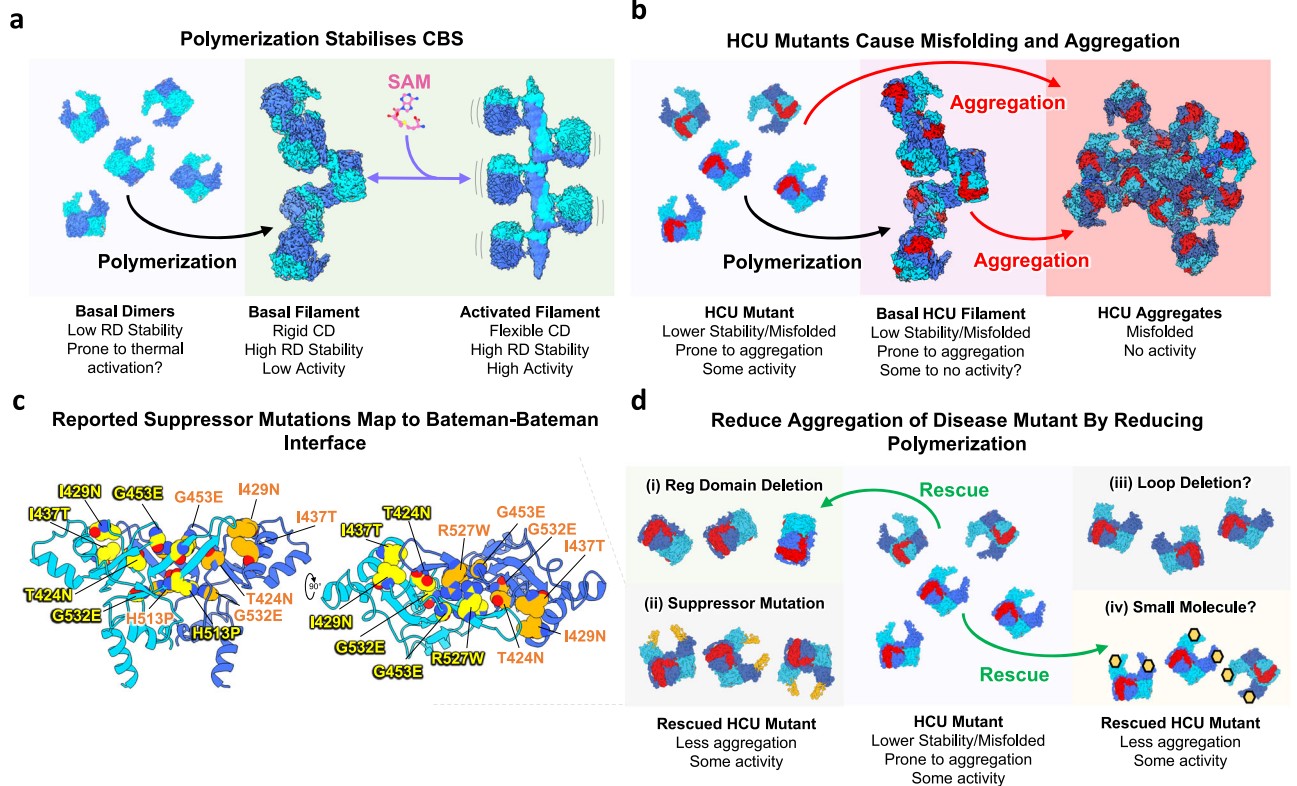

**Fig. 6 | Filament formation stabilises CBS but potentially increases the aggregation of disease mutants. a** Proposed model of CBS stabilisation by polymerisation and allosteric activation. **b** Proposed model of HCU mutant CBS polymerisation and its relation to CBS aggregation. Red indicates mutation on the catalytic domain. **c** Previously reported HCU mutant suppressor mutations are located at the Bateman-Bateman interface. **d** Proposed model of HCU mutant

rescue by reducing polymerisation and aggregation. Rescue has been shown by (i) deletion or (ii) mutation of the regulatory domain, which may sacrificially reduce CBS's natural propensity to polymerise into filaments. The (iii) deletion of the loop 515–526 or (iv) binding of a small molecule at the Bateman-Bateman interface could also reduce polymerisation and reduce aggregation of HCU mutants.

depleted (Fig. 5). Altogether, this not only demonstrates human CBS filamentation *in cellulo*, but also provides a functional context that both *S. cerevisiae* Cys4p and human CBS oligomerise under certain metabolic state. CBS catalyses the first committed step of the trans-sulfuration pathway that digresses from the methionine cycle and depletes the methionine pool. Our data reveals a feedback mechanism regulating one-carbon flow between the methionine and transsulfuration pathways in response to cellular methionine levels. Beyond the *S. cerevisiae* and human enzymes, AlphaFold predictions[39] and sequence alignment of Bateman modules from various CBS orthologues suggest that filament formation is possibly conserved in chordates (Supplementary Figs. 18, 19). This therefore implies that CBS oligomerisation could be an evolutionary strategy overall but could involve different structural components and serve different functional outcomes.

Future studies are clearly warranted to elucidate in detail how CBS filamentation equips the cell in times of metabolic growth/stress. For many metabolic enzymes, filamentation and oligomerisation serve to provide new ligand binding sites, generate unique catalytic

conformations, or transduce signals across multiple enzyme subunits. Such functional modification to the enzyme is often reflected in a significant increase (or decrease) in cooperativity and activity upon filament formation[31,32]. For the case of human CBS, we observe only a modest increase in cooperativity in comparison to the loop deleted CBS[Δ516–525] construct (Fig. 4a), which falls short of statistical significance from our replicates. This is not surprising as the CBS filament interface involves a single point of contact (Figs. 1, 2). In other enzymes, such as IMPDH, which forms a filament of tetramers, multiple contacts are formed across single filament interfaces with corresponding high values of cooperativity[40]. We do, however, find that the full-length CBS filament is more stable and less prone to thermal activation than the loop-deleted CBS[Δ516–525] dimer (Fig. 4b, c). We previously observed that the CBS[Δ516–525] construct is conformationally flexible without SAM, presenting as two populations in ion mobility experiments[17]. Thus, we propose that the primary objective of filament formation in CBS is to increase the kinetic stability of the regulatory domain to maintain the basal conformation of the enzyme[28,29,35] (Fig. 6a). The notion of filament formation increasing stability is further supported by the His-

 

tagged construct that presents longer polymers, higher stability, and less degradation than the construct without a His-tag (Fig. 4b, Supplementary Fig. 2). Increased stability and activity of human CBS due to a C-terminal His-tag has been previously reported[20]. Additionally, we found that the His-tag affects the entropy and enthalpy of the first SAM binding event in ITC in comparison to the constructs without a His-tag (Supplementary Figs. 14b and 15). Unfortunately, we cannot structurally rationalise this effect as we observed no interpretable density in the maps of this construct for the affinity tag at the Bateman-Bateman interface (Figs. 1, 2). We theorise though, that the His-tag could be acting as a proxy for a yet unidentified ligand that may regulate CBS oligomerisation (as seen in our *in cellulo* studies) or affect the putative cryptic SAM binding site. Further investigations are clearly warranted; however, these observations show that filament formation can be modulated (positively or negatively) by changes at the Bateman–Bateman interface.

Inherited mutations in CBS result in classical homocystinuria (HCU), in which most recorded mutations are missense[14], and the dominant molecular mechanisms have been recognised as protein misfolding and aggregation[8–10] (Fig. 6b). Rescue of mutant CBS activity has been documented by chemical chaperones[34,41], haem arginate[42], and proteostasis inhibitors[43], suggesting a small molecule therapy could be developed[44,45]. It is intriguing that genetic suppression in a yeast model of the disease has also been reported, where deletion[26] or missense mutations on the regulatory domain[25] can overcome the deleterious effects of some HCU mutations. Disease-associated mutants, in general, can produce hydrophobic patches in the protein due to local misfolding, which will result in aggregation[46–48]. It is probable that many HCU mutations generate hydrophobic patches on the catalytic domain resulting in further non-specific interactions that lead to aggregation[9] (Fig. 6b).

Considering our findings, we hypothesise that a mechanism of rescue could be the reduction of the natural propensity for CBS to polymerise which could reduce one pathway towards aggregation (Fig. 6d). In support of our reasoning, (1) deletion of the regulatory domain prevents filament formation (Supplementary Fig. 2) and (2) the seven reported suppressor mutations from the yeast model of disease can be all mapped onto the hydrophobic face of the regulatory domain that forms the Bateman–Bateman interface (Fig. 6c, Supplementary Fig. 20). All seven residues are conserved in chordate CBS (Supplementary Fig. 18) and are predicted to alter interactions at the oligomeric interface (Supplementary Fig. 20b). The original report suggested that these mutants altered the interaction between the regulatory and catalytic domains trapping CBS in a partially open conformation which is non-responsive to SAM. However, in the background of the wild-type enzyme no slight increase in basal activity was observed with these mutations[49], and as such, we believe that altered oligomerisation should be considered as an aspect of rescue. As dimeric CBS$^{\Delta516-525}$ behaves almost like full-length (i.e., is active and SAM responsive), a small molecule that disrupts CBS polymerisation and reduces aggregation could be a potential therapeutic avenue for the treatment of HCU (Fig. 6d).

Overall, we have determined multiple structures of human CBS, showing that the full-length enzyme polymerises as an active filament that changes conformation due to SAM. CBS polymerisation is further observed in cell-based fluorescence microscopy, reinforcing the necessity of the RD loop and response to the nutrient state. Future work should consider further the role of CBS polymerisation in protein misfolding and aggregation. It is interesting to consider that there are catalytic domain HCU mutations that result in a CBS enzyme with normal basal activity but non-responsive to SAM[41]. SAM non-responsive HCU mutations in the regulatory domain exhibited an enzymatic activity closer to the activated state[50]. These findings suggest that some mutants may lock CBS in one conformation. Cryo-EM

studies of these and other disease associated mutants will give insight into the molecular mechanism of protein misfolding of CBS and may have implications in understanding the misfolding of other multi-domain metabolic enzymes.

## Methods

### Cloning, expression, and purification of human CBS proteins

The gene for human CBS (UniProt P35520) was cloned into pNIC-Bsa4 and pNIC-CH encoding for a TEV cleavable N-terminal and permanent C-terminal His-tag, respectively. The constructs CBS$^{\Delta516-525}$ and CBS$^{CD}$ (residues 1–413), along with single point mutations of CBS, were introduced using In-Fusion (Takara) or QuikChange (Agilent) mutagenesis and confirmed by sequencing. CBS was expressed in *E. coli* Rosetta (DE3) cells in auto-induction Terrific Broth (TB) supplemented with 50 μg/ml kanamycin, 34 μg/ml chloramphenicol, 0.3 mM δ-aminolevulinic acid, 0.0025% pyridoxine–HCl, 0.001% thiamine–HCl, and 0.1 mM ferric chloride at 30 °C, 200 rpm for 24 h. Cells were resuspended in lysis buffer (50 mM sodium phosphate, pH 7.5, 500 mM NaCl, 0.5 mM TCEP, 5% glycerol, 1.0% Triton X-100, 0.1 mM PLP, 2 mg/ml lysozyme) and lysed by sonication. CBS proteins with a TEV cleavable N-terminal His-tag were purified using Ni-NTA agarose (Qiagen) resin and were treated to gel filtration using a Superose 6 Increase 16/600 column or Superdex 200 Hiload 16/600 column (Cytiva) equilibrated in storage buffer (25 mM HEPES, pH 7.5, 500 mM NaCl, 0.5 mM TCEP, 5% glycerol). Fractions containing CBS protein were pooled and treated with His-tagged TEV protease overnight at 4 °C and then passed over Ni-NTA agarose resin to remove the TEV protease and uncleaved protein. CBS proteins with a permanent C-terminal His-tag were purified using TALON (Clontech) resin, followed by anion exchange using a Hitrap Q column (Cytiva). Anion exchange elutions were polished by gel filtration using a Superose 6 Hiload 16/600 column (Cytiva) equilibrated in storage buffer (25 mM HEPES, pH 7.5, 500 mM NaCl, 0.5 mM TCEP, 5% glycerol). For all purifications appropriate fractions were pooled, concentrated to 5-20 mg/ml, snap frozen, and stored at −80 °C.

### Clear and blue native-PAGE

Clear and blue native-PAGE was carried out according to the manufacturer's instructions (Life Technologies). CBS constructs were diluted in 25 mM HEPES, pH 7.5, 200 mM NaCl, and 2.0 mM TCEP. In some cases, 1 mM SAM was added and incubated at room temperature for 5 min before loading. All samples were at 1.0 mg/ml and 8.0 μg total protein.

### Analytical size exclusion chromatography (SEC)

Analytical SEC was carried out using a 10/300 GL Superose 6 Increase column (Cytiva) equilibrated in 25 mM HEPES, pH 7.5, 500 mM NaCl, 0.5 mM TCEP, and 5% glycerol. 250 μl of each CBS construct was loaded at 3.0 mg/ml with a flow rate of 0.3 ml/min. 500 μl fractions were collected for analysis by SDS−PAGE. Gel filtration standards were purchased from Bio-Rad. Approximate molecular weights of CBS oligomers were calculated using the relationship between the log of the known molecular weight of the standards and their respective partition coefficient, $K_{av}$.

### Grid preparation and cryo-electron microscopy

CBS$^{FL}$ and CBS$^{FL-CHis}$ were diluted to 1.0 mg/ml (15 μM) into 25 mM HEPES, pH 7.5, 200 mM NaCl, 2.0 mM TCEP, 0.005% (v/v) tween-20 for the basal state. For the activated state, CBS$^{FL-CHis}$ was diluted to 0.75 mg/ml (11.25 μM) into 25 mM HEPES, pH 7.5, 200 mM NaCl, 2.0 mM TCEP, 0.005% (v/v) tween-20, 5 mM SAM. Grids were prepared using a FEI Vitrobot Mark III (Thermo Fisher Scientific) at 4 °C and 100% humidity. 3 μl of sample was applied to a plasma treated gold coated R 1.2/1.3 300 mesh holey carbon grid (Quantifoil), with a blot force of 0, a blot time of 3 s, and a wait time of 10 s.

 

Movies of the CBS$^{FL-CHis}$ basal state were collected at eBIC (Diamond Light Source) on a Titan Krios equipped with a Falcon 3EC direct electron detector (Thermo Fisher Scientific) operating in counting mode. Images were imaged at 300 kV with a magnification of ×75,000, corresponding to a pixel size of 1.085 Å. 40 frames over 60 s were recorded with a defocus range of −0.9 μm to −3.0 μm with a total dose of 37.85 e$^-$ A$^{-2}$ (0.823 e$^-$ A$^{-2}$ per frame). A total of 1740 movies were collected in a single session. MotionCor2[51] was used to correct beam-induced motion and CTF was estimated using CTFFIND-4.1[52]. For helical reconstruction, particles were picked using the filament picker of Relion 3.0.8[53] resulting in 239,739 particles extracted. Helical picks were subjected to multiple rounds of 2D classification, producing 82,810 particles that were imported to CryoSPARC-3.1.0[54]. Further 2D classification to remove any junk particles reduced this to 76,663 particles. Helical parameters were roughly determined from a low-resolution Glacios collected map. Non-uniform helical refinement with D1 symmetry (C2 symmetry perpendicular to the helical axis) imposed resulted in a 3.7 Å map with a refined helical twist of −108.4° and a rise of 51.2 Å. For single particle analysis, particles were auto-picked using the Relion 3.0.8[53] (Laplacian of Gaussian function), resulting in 760,869 particles extracted. One round of 3D classification with 4×-binned images and a model from a subset of the data was used to remove bad particles and contamination. This resulted in 392,163 particles that were unbinned and subjected to per-particle CTF refinement and Bayesian polishing. After another round of masked 3D classification, one class consisting of 207,509 particles was identified to have the highest level of structural detail. A further round of CTF refinement and Bayesian polishing followed by masked auto-refining was used to produce particles for cryoSPARC-3.1.0[54]. 2D classification followed by heterogeneous refinement reduced the number of good particles to 188,230. Multiple rounds of non-uniform refinement, local CTF refinement, and local non-uniform refinement with C2 symmetry imposed resulted in a 3.0 Å map.

EER formatted movies of the CBS$^{FL}$ basal state were collected at the York Structural Biology Laboratory (YSBL) on Glacios equipped with a Falcon 4 direct electron detector (Thermo Fisher Scientific). Images were imaged at 200 kV with a magnification of ×150,000, corresponding to a pixel size of 0.934 Å. Movies over 5.18 s were recorded with a defocus range of −1.4 to −2.0 μm with a total dose of 50 e$^-$ A$^{-2}$. A total of 2628 movies were collected in a single session. All movies were imported into cryoSPARC-3.3.2[54] where they were subjected to patch CTF estimation and patch motion correction. For helical reconstruction, particles were picked using the filament tracer resulting in 749,213 particles extracted. Multiple rounds of 2D classification to remove any junk particles reduced this to 98,993 particles. Particle curation based on CTF fit further reduced this to 89,761 particles. Initial helical parameters were determined from the CBS$^{FL-CHis}$ map. Non-uniform helical refinement with D1 symmetry imposed resulted in a 3.9 Å map with a refined helical twist of −108° and a rise of 51 Å. For single particle analysis, 1,190,611 particles were picked and extracted using template-based picking. Rounds of 2D classification resulted in 160,400 particles that were subjected to ab initio reconstruction and heterogeneous refinement with three classes. Two classes were further separately processed using local non-uniform refinement with C2 symmetry imposed, resulting in 3.8 Å maps of both partially degraded and non-degraded CBS.

Movies of the CBS$^{FL-CHis}$ activated state (SAM-bound) were collected at eBIC (Diamond Light Source) on a Titan Krios (Thermo Fisher Scientific) equipped with a K3 (Gatan) direct electron detector operating in super-resolution mode. Images were imaged at 300 kV with a magnification of ×81,000, corresponding to a pixel size of 0.53 Å. 44 frames over 3.53 s were recorded with a defocus range of −0.9 to −3.0 μm with a total dose of 39.96 e$^-$ A$^{-2}$ (0.908 e$^-$ A$^{-2}$ per frame). 11,220 movies were collected in a single session. All movies were imported into cryoSPARC-3.1.0[54] where they were motion corrected, and the CTF was estimated using patch motion correction (Fourier cropped to

1.06 Å) and patch CTF estimation, respectively. Processing the activated state took considerable effort, and initially, filaments were picked using the filament tracer without templates on 2790 micrographs. The resulting best classes from 2D classification were then used for another round of filament picking. Another round of 2D classification and picking the best classes were used to produce templates representative of the two dominant views with different filament widths. Two separate rounds of filament picking on the entire dataset resulted in 2,374,969 and 2,865,445 particles, which were eventually merged into a pool of 492,224 particles after many rounds of 2D classification and removal of duplicate particles. Helical parameters were roughly determined by visual inspection of a previously determined low-resolution Glacios collected map. Rounds of non-uniform helical refinement with D1 symmetry imposed were used to re-centre particles and remove duplicates within 44 Å resulting in 425,260 particles. One further round of non-uniform helical refinement with D1 symmetry imposed resulted in a map at 4.0 Å resolution of the full filament with a refined helical twist of −178.6° and a rise of 46.7 Å. Though this map had good features for the central portion of the filament, the protruding catalytic domains have blurred features due to relative flexibility. Subsequently, this map was used to make a soft mask (10 Å dilation with a soft padding of 50 Å) of the central filament region. Imposing this as a static mask during non-uniform helical refinement with D1 symmetry imposed resulted in a 4.1 Å resolution of the central regulatory domain with a helical twist of −177.7° and a rise of 47.2 Å. To improve the resolution of the central regulatory domain, the particles were then subjected to a masked local refinement with D1 symmetry applied and a less soft central regulatory domain mask (10 Å dilation with a soft padding of 20 Å). This resulted in a map with a resolution of 4.1 Å. Though distortions due to the flexibility of this central region were apparent at the edges of the map, the three dimeric repeats at the centre of the map had improved features. To improve the quality and resolution of the highly flexible catalytic domains, masked local refinement with D1 symmetry imposed with a soft mask of the most central catalytic and regulatory domains (6 Å dilation with a soft padding of 20 Å) was used. Here the global helical map was filtered to 20 Å and alignments only considered a resolution of up to 9 Å, resulting in a map of 8.3 Å resolution.

## Model fitting, refinement, and validation

For the CBS$^{FL}$ basal state structures, initially, three CBS$^{\Delta516−525}$ structures (PDB: 4COO) were fitted using Molrep[55] and the missing loop 513–527 was manually built in Coot[56]. Multiple copies of the CBS$^{FL}$ model were docked into each map as appropriate using Phenix[57]. Rounds of refinement in Phenix[57] and flexible fitting using Isolde[58] were then used to refine the structure with manual adjustments in Coot[56]. For the activated state, multiple copies of the isolated Bateman domain from our basal structure were docked using Phenix[57] into the central locally refined map and then flexibly fitted into a 10 Å filtered map using Namdinator[59]. This was followed by a second round of flexible fitting using the non-filtered map. The crystal structure of the Bateman loop-deleted dimer bound to SAM (PDB: 4UUU) was used as a guide to dock SAM into the appropriate density. This model and our basal state model were then used as references for refinement of the SAM-bound model using Isolde[58] and Phenix[57]. Additional modelling of the flexible catalytic domains was done by manually fitting the catalytic domain structure from PDB 4PCU, using ChimeraX[60], into the 8.3 Å map of a single catalytic domain. Multiple copies of the resulting catalytic domain model were then manually docked into the global helical refined 4.0 Å SAM-bound map using ChimeraX[60]. All models were validated using Molprobity[61].

## Enzyme activity assay

Kinetic parameters were determined by monitoring hydrogen sulfide (H$_2$S) production using the fluorescent probe 7-azido-4-methylcoumarin (AzMC)[62]. Assays were performed in 25 mM HEPES,

pH 7.5, 200 mM NaCl, 5 µM PLP, 10 mM glutathione and 10 µM AzMC, with 0.01% triton-x 100, in 384-well black plates, as a final assay volume of 50 µl. A final concentration of 100 nM of each CBS construct was used. For Michaelis–Menten kinetics, cysteine was varied 0-40 mM with a constant 10 mM homocysteine, and homocysteine varied 0–10 mM with a constant 40 mM cysteine. 300 µM SAM was added when appropriate. SAM titration assays were performed with a final concentration of 10 mM homocysteine and 40 mM cysteine, and SAM was added at a range of 0–1.0 mM. Thermal activation was carried out with CBS protein at 1 µM in 25 mM HEPES, pH 7.5, 200 mM NaCl and 50 µM PLP as 50 µl aliquots treated at different temperatures using a VeritiPro thermal cycler (Thermo Fisher Scientific) for 2 min. Treated samples were then put into ice before activity was assayed with 10 mM homocysteine and 40 mM cysteine. All plates were preincubated with enzyme for 10 min at 37 °C before the addition of cysteine. Plates were sealed and spun at $900 \times g$ for one minute before loading into the plate reader. $H_2S$ production was monitored by fluorescence at 450 nm ($\lambda_{ex} = 365$ nm) using an OmegaSTAR (BMG Biotech) at 37 °C. Each plate was read for one hour with a reading every one minute and raw rates were determined using MARS software (BMG Biotech). Activity readings were calibrated using a standard curve of known $H_2S$ concentrations using sodium hydrosulfide hydrate as an $H_2S$ source. Kinetic analyses were done in GraphPad Prism.

## Thermal shift assay
CBS constructs were diluted in thermal shift buffer (25 mM HEPES, pH 7.5, 200 mM NaCl, 2.0 mM TCEP) to 0.3 mg/ml with SYPRO-Orange (Invitrogen) diluted 1000X in a total volume of 20 µl. A QuantStudio 5 RT-PCR machine (Thermo Fisher Scientific) was used to measure melting temperatures.

## Isothermal titration calorimetry
Purified CBS proteins were buffer exchanged into 20 mM HEPES, pH 7.4 using Zeba spin columns (Thermo Fisher Scientific) at 4 °C. To prevent precipitation CBS$^{FL-CHis}$ and its mutants were buffer exchanged into 20 mM HEPES, pH 7.4, 0.01% triton X-100 whereas CBS$^{RD}$ was exchanged into 20 mM HEPES, pH 7.4, 200 mM NaCl. The appropriate buffer was then used to dissolve SAM from stocks to 500 µM. A MicroCal PEAQ-ITC machine with v1.21 control software for data collection (Malvern Panalytical) was used to perform ITC. CBS constructs were tested at 30-60 µM monomer in a 200 µl sample cell and were injected with 0.4 µl followed by $44 \times 0.8$ µl of SAM with 150 s spacing at 25 °C. Heats of dilution were determined by separate runs of SAM injected into the buffer alone. Integrated heats were fitted using the Microcal PEAQ-ITC analysis software v1.30 (Malvern Panalytical) to obtain $n$, $K_d$, $\Delta H$, and $-T\Delta S$.

## Cell lines and plasmids
MCF7 (ATCC no. HTB-22), MDA-MB-231 (ATCC no. HTB-26), BT549 (ATCC no. HTB-122) and PC-3 cells (ATCC no. CRL-1435) were cultivated in RPMI 1640 medium (Sigma-Aldrich) supplemented with 10% foetal bovine serum (FBS, Vitrocell). NIH3T3 (ATCC no. CRL-1658), mouse embryonic fibroblasts (MEFs) of C57BL/6 strain (donated by Â. Saito) and human fibroblasts (hFB) (ATCC no. CRL-2703) were cultured in high-glucose Dulbecco's modified Eagle medium (DMEM, Sigma-Aldrich) supplemented with 10% FBS (Vitrocell). Human CBS was cloned into pcDNA5-FRT encoding for a C-terminal Flag-mKO2-tag for the mammalian expression vectors. The constructs CBS$^{\Delta516-525}$, along with single point mutations of CBS, were amplified from the previously generated sequences, introduced into the pcDNA5-FRT backbone using In-Fusion (Takara) and confirmed by sequencing.

## Transfection and treatments
To screen for CBS aggregates in cells, $8 \times 10^3$ cells per well were plated into CellCarrier™ 96-well microplates (PerkinElmer, Germany,

#6055300). The next day, cells were transfected with 0.1 µg plasmid per well using Lipofectamine 2000 (Thermo Fisher Scientific). 24 h after transfection, cells were washed with 1xPBS and incubated with different media conditions for 8 h. RPMI-1640 Medium without L-methionine, L-cystine, and L-glutamine (Sigma-Aldrich, #R7513) was supplemented with 10% dialysed foetal bovine serum (Sigma-Aldrich, #F0392) and the specific amino acids (Sigma-Aldrich): 0.1 mM L-methionine (#M5308), 0.2 mM L-cystine (#C8755), and 4 mM L-glutamine (#G5792). SAM (New England Biolabs, #B9003S) was added at a concentration of 320 µM, and all the conditions were treated with the same amount of solvent (50 µM of $H_2SO_4$ and 0.1% ethanol). Cells were then fixed with 2% PFA for 10 min, and the nuclei were labelled with DAPI in PBS 0.1% Triton for 10 min. After nuclear labelling, cells were washed with 1xPBS, and the plates were stored with 100 µl of PBS/well, protected from light at 4 °C or taken directly to the Operetta™ automatic cell imaging system (PerkinElmer, MA, USA). For data storage, we used Columbus software v2.4.0 (PerkinElmer, MA, USA).

## Image analysis
Exported images were organised following the Columbus export format. All subsequent image processing and analysis were conducted using ImageJ software. Cells were identified based on their nuclear morphology, which was enhanced by combining nuclear and transfection channel images. This process began with applying a Gaussian blur (sigma = 3) to these combined images, aiding in the more precise visualisation of cellular structures. The individual puncta or filaments, characterised by mKO$_2$ fluorescence, were processed for enhanced detection. Initially, a consistent contrast adjustment was applied across all images. This step was followed by applying the Enhance Local Contrast (CLAHE) algorithm to further improve the visibility of puncta and filaments. Subsequently, images were binarised using a suitable thresholding technique to identify these structures distinctly. For each identified object (puncta or filament), the fractal dimension ($D$) value was calculated using the Fractal Box Count method. This metric provided a quantitative measure of the complexity of the morphology of the detected structures. A $D$-threshold of 0.7 was set to categorise the cells based on the morphological complexity of the objects within their expanded cytoplasm. Cells were classified into either 'filamented' or 'punctuated' categories, depending on whether most objects within the expanded cytoplasm met or exceeded the $D$-threshold. In addition to fractal analysis, the perimeter length of each object was measured. A threshold of 10 micrometers was employed to distinguish between short and long perimeters. This measurement provided further insight into the morphological characteristics of the cellular structures observed. In both scenarios, ties were disregarded from the total percentage of cells.

## Structural analysis using AlphaFold multimer
CBS sequences were obtained from UniProt[63], and their sequences aligned using Clustal Omega[64]. AlphaFold2 multimer[65] was used through the implementation in ChimeraX[60]. Six copies of the selected CBS Bateman domains were used as the input and the top-scored model was used for structural analysis. All lower-scored models were essentially identical to the top scorer for all orthologues.

## Reporting summary
Further information on research design is available in the Nature Portfolio Reporting Summary linked to this article.

# Data availability
The authors declare that the main data supporting the findings of this study are available within the article and Supplementary Information. EM maps and models generated in this study, of CBS basal state (EMD-19735, PDB 8S5H, EMD-19736, PDB 8S5I, EMD-19737, PDB 8S5J, EMD-19738, PDB 8S5K), degraded CBS tetramer (EMD-19739, PDB 8S5L) and

CBS + SAM activated state (EMD-19740, PDB 8S5M, EMD-19741, EMD-19742), have been deposited to the Electron Microscopy Data Bank (EMDB) and Protein Data Bank (PDB). Other structures referenced in this article are indicated, including PDB ID 4COO, 4UUU, and 4PCU. Source data are provided with this paper.

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

## Acknowledgements

We thank all members of the SGC Oxford Biotechnology team for their molecular biology support. We thank the Oxford Particle Imaging Centre (OPIC) electron microscopy facility for grid screening and data collection. We acknowledge the Diamond Light Source for access and support to the UK's Electron Bio-imaging Centre (eBIC, under BAG proposal EM20223) funded by the Wellcome Trust, MRC, and BBRSC. We specifically want to thank James Gilchrist and Yuriy Chaban for their assistance. We also want to thank Laura Diez-Siez for her help in the initial EM screening and data collection. Additionally, we thank Johan Turkenburg, Sam Hart, and Jamie Blaza for their assistance in collecting Glacios data at York Structural Biology Laboratory (YSBL). The YSBL is funded by the BBSRC, the Wellcome Trust (grant number 206161/Z/17/Z), Tony Wild, and the Wolfson Foundation. A.L.B.A., S.M.G.D. and R.A.C.M. are funded by FAPESP (grant numbers 2021/05726-6 and 2023/01388-4). We acknowledge Dr. Angela Saito from LNBio/CNPEM for generously donating the MEF cells used in this work. We also wish to thank Brian Marsden and Chris Sluman for their bioinformatics support. The Structural Genomics Consortium is a registered charity (Number 1097737) that receives funds from AbbVie, Bayer Pharma AG, Boehringer Ingelheim, Canada Foundation for Innovation, Eshelman Institute for Innovation, Genome Canada, Innovative Medicines Initiative (EU/EFPIA) [ULTRA-DD grant no. 115766], Janssen, Merck & Co., Novartis Pharma AG, Ontario Ministry of Economic Development and Innovation, Pfizer, São Paulo Research Foundation-FAPESP, Takeda, and Wellcome Trust (092809/Z/10/Z). Initial work on this project was funded by a Wellcome Trust Pathfinder Award to W.W.Y. T.J.M. received cryo-EM training through the Wellcome/MRC funded programme (218785/Z/19/Z), was a recipient of a Pump-Priming Award from the Medical Sciences Division, University of Oxford, and is funded by an HCU Network North America and Australia Research grant.

## Author contributions

T.J.M. and W.W.Y. designed the experiments. T.J.M. expressed and purified CBS constructs, carried out biochemical experiments, screened and collected EM data, analysed, and refined all CBS structures. H.J.B. did the initial screening and collected EM data. J.T. and D.S.M.F. did EM screening. C.S.D. carried out initial CBS construct cloning, expression testing, and optimisation. A.B. assisted in the EM processing of the CBS^FL basal state and deposition of data. D.A., R.A.C.M., A.L.B.A., and S.M.G.D. designed, performed, and analysed the cell biology experiments. T.J.M. and W.W.Y. carried out data analysis and wrote the manuscript with contributions from all authors.

## Competing interests

The authors declare no competing interests.
