## [Peer Review File · Nature Communications]

Reviewers' Comments:

Reviewer #1:

Remarks to the Author:

This is an exciting report of the filamentous structure of an important metabolic enzyme, human CBS. Prior reports suggested some higher-order structure formation by CBS, though most high-resolution studies relied on a construct with a segment of the enzyme deleted. Those studies concluded that the enzyme was tetrameric, however, the deleted segment was found in the current work to be crucial for filament (polymer) formation. The authors determine the structures of the filamentous form of CBS in a basal state, as well as a state activated by binding the allosteric activator SAM. They find very large, striking conformational changes in the filament, and these conformational changes at the tetrameric level are perfectly consistent with the prior x-ray crystallographic reports. The authors' most important finding, beyond the highly significant result of the filamentous nature of the enzyme, was that filamentation increases the stability of the enzyme (to thermal and perhaps protease degradation). A secondary finding was that the filamentation increased the cooperativity of the enzyme, though the statistical significance of the differences in Hill coefficients should be interrogated more rigorously. The authors also contribute to the discussion of the number of SAM binding sites per CBS chain (1 or 2). They find 2 binding events using ITC, but argue that the second, occurring at much higher concentrations of SAM than the first (600 nM vs 160 nM), is actually due to the conformational change within the filament, and not actual SAM binding. They provide evidence in the form of mutant and deletion constructs to support this argument, though alternative hypotheses are not ruled out and their interpretation of the ITC could be tested further using ITC itself. Finally, the authors make interesting arguments regarding the role of polymerization of CBS in the disease-causing mutations, namely in increasing the propensity for aggregation of misfolded forms, and they propose disruption of polymerization as a new therapeutic approach to treat these diseases.

Below I itemize comments made while reading the manuscript that may help the authors in improving readability:

1. In the Abstract. "Human CBS appends to its conserved catalytic domain a regulatory domain". I understood this sentence the second time I read it, but the first time I read it, it was confusing and sounded like the enzyme CBS had a catalytic activity that appended domains onto itself. Something like "Appended to the catalytic domain is a regulatory domain in human CBS" or more simply, "Human CBS has an additional regulatory domain appended to its catalytic domain", or something similar.
2. Same sentence, which ends with "... that modulates activity by S-adenosyl-L-methionine (SAM) and promotes oligomerization, however, the molecular basis is unknown". The molecular basis of what? I assume this means either the molecular basis of regulation, or of oligomerization, or both, but the sentence could be written in a way that is more clear.
3. Last sentence of the abstract, "elaborate our understanding" is a strange use of the word elaborate. I would reword.
4. Introduction, "lost-of-function" should probably be "loss-of-function".
5. Extended Figure 1 A and text on page 2 (lines 58-69), it is hinted by Extended Figure 1A, but not stated anywhere, that there might be two active sites per dimer (Extended Figure 1A shows that the active site of the blue chain is blocked when SAM is not bound, and then when SAM is bound, the active site of shown open, but in the symmetry-related subunit (suggesting two active sites, though it is unclear).
6. Extended Figure 2b, the colors are difficult to distinguish (in a printout, but not if read online).
7. Line 98 "Here the RD sits atop the active site entrance of the catalytic domain hindering substrate access (Fig. 1c,e)". Figure 1c doesn't identify the active site entrance. Perhaps the authors meant Fig. 1f and e?
8. The blue colors in Figure 1e, notably in the text labels, are very difficult to distinguish. They are easily distinguishable in the map as shown here, but not the labels. The blues are more difficult to distinguish in Extended data Fig. 3d-e. This is especially true when in printed form.

9. In Figure 1d, the legend indicates that one subunit is colored as in Figure 1a, however its partner subunit in the dimer is colored white or grey I think. This should be made clear because then the other two dimers shown (2 and 4) are colored as in Figure 1c, although Figure 1c should indicate what the colors mean (the two subunits of the domain-swapped dimer probably?).
10. Colors in extended figure 1 should be defined (the two blue colors in b-f). These color choices make it harder to see the difference between the two subunits. The boxed regions in extended data Figure 3b should be connected somehow to the regions shown in d-e to make it clear that the closeup views map to those locations in the zoomed-out view.
11. Line 182, the reference to Fig. 2d, the next mentions loop 516-525, and residues 422-426 of alpha-helix 15, but these aren't identified in Fig. 2d.
12. line 186, "little clashes" might be better as "fewer clashes".
13. Fig. 3a, they might mention that the dimer interface is being shown, with one dimer colored white and one salmon, hence why two molecules of SAM are shown.
15. In the paragraph beginning at line 233, references to Extended Data Fig. 6a should be to 7a, and Fig. 3a should be Fig. 4a. Perhaps the extended data figures need to be reordered so that they are called out in order?
16. Are the differences in Hill coefficients statistically significant? The error bars lead these values to overlap. Some other statistical tests could be applied. This could be an important point in the paper, that the filament increases the cooperativity of the enzyme.
17. The paragraph beginning on line 249, references to extended data Fig. 5a should be to 6a. also, it would be really difficult to read this figure in printed form due to the same size of the text fonts.
18. In the results and discussion, the authors argue that the second observed binding of SAM at 600 nM is actually not a binding event, but heat dissipation from the conformational change of the enzyme in the filament. Why should it occur at this exact concentration of SAM every time? Is that a function of how the ITC was done (the authors argue that it takes time for the conformational change to occur after SAM binding)? If titrations were done that did not reach 600 nM SAM, would this heat dissipation still occur as expected? If the titration were slowed, would the heat dissipation occur at a different concentration of SAM?
19. The authors show that mutation of the first SAM binding site eliminates both binding events, and that the construct with only the RD (and no regulatory loop) has only 1 and argue that these results are evidence that the second so-called binding event is actually a conformational change. Could it not also be that the second binding site is not in the residues in the RD construct, but does overlap the first SAM binding site such that the mutations affect its binding as well?
20. The ITC results could be presented and discussed more to show for one thing how strong the data are for the two binding events and how to interpret them. Also, why are the enthalpies and entropies so different for CBSFL and CBS-FL-Chis with respect to the first SAM binding site (Ext. Fig. 6d)? What are the measures of quality of fitting of the ITC data? I assume the red line in the delta H vs. Molar Ratio is the fit, while the points are the experimental data? This should be explained in the legend.
20. Fig. 5a, is another figure with very small fonts making it difficult to read, especially when in print without the option to zoom.

Reviewer #2:

Remarks to the Author:

In the manuscript "Architecture and regulation of filamentous human cystathionine beta-synthase," McCorvie et al. present a comprehensive study of the quinary structure of the human CBS. They expressed, purified, and characterized five CBS protomers (wild type and terminally His tag-flagged wild type CBS, catalytic domain, regulatory domain, and CBS with the deleted regulatory region of 516-525 amino acid residues) using many biophysical, biochemical, and cryo-

EM imaging methods with subsequent data processing techniques. The significant findings are a/the demonstration of filament formation of the wild type CBS enzyme, b/ changes in quinary structure with the increased catalytic activity of the filaments in the presence of S-adenosylmethionine c/the possible implications for novel therapeutic targets in the SAM S2 binding site. In summary, the study brings novel data on an additional layer of regulation of CBS activity with translational potential.

Due to limited expertise in structural biology, the reviewer cannot critically assess the methods, results, and bioinformatic interpretation of CBS quinary structure. Here are several comments to consider:

1. Relevance of in vitro filament formation for in vivo biology. The study used purified CBS protomers to demonstrate the formation of CBS fibrils under non-physiological conditions. The critical question is whether CBS fibrils also form in vivo in human cells. The authors provided only one indirect supporting indication for human CBS filamentation in the cellular environment, i.e., the formation of punctate foci of the yeast CBS ortholog (encoded by the *CYS4* gene) during the stationary phase. It is unlikely that the present study will be expanded experimentally to prove the filamentation of human CBS in cellular systems; however, the authors should discuss future directions to confirm the filamentation of wild type human CBS in vivo.

2. Size of oligomers and length of filaments. The study shows that the filaments are assembled from dimers. The estimated size of the dimer is about 120 kDa. Can the authors comment on the differences between the predicted molecular weight of the dimer and the apparent size on Superose gel filtration (major peak size at 15 ml, i.e., between 158 and 670 kDa), clear native gel (major signal between 242 and 480 kDa) and BN gel (major signal between 146 and 242 kDa)? In addition, I could not find information on the lengths of fibrils and the underlying number of subunits; the authors may consider adding this information.

3. Pathogenicity of the p.D444N variant. The authors show in Figure 3 the model of the SAM S2 binding site also containing a well-studied pathogenic mutation p.D444N. The pathogenicity of this variant is intriguing as it is constitutionally active to overactive and cannot be further activated by physiologically relevant SAM concentrations (e.g. reference 34, reference Kluijtmans et al. Defective cystathionine beta-synthase regulation by S-adenosylmethionine in a partially pyridoxine responsive homocystinuria patient. Clin Invest. 1996 Jul 15;98(2):285-9 and reference Evande et al. Alleviation of intrasteric inhibition by the pathogenic activation domain mutation, D444N, in human cystathionine beta-synthase. Biochemistry 2002 Oct 1;41(39):11832-7). Using their model, can the authors shed light on the possible pathogenic mechanism(s) of the p.D444N mutation?

4. Suppression mutations. The authors state: "It is intriguing that genetic suppression in a yeast model of the disease has also been reported, where deletion²⁹ or missense mutations on the regulatory domain²⁹ can overcome the deleterious effects of the most common HCU mutations." The reference for suppression by missense mutation seems incorrect and should be replaced by reference 28. Moreover, the statement is incorrect as reference 28 studied functional correction of only two mutants (p.I278T and p.V168M). Please, revise accordingly.

5. Sulfur spelling. The authors use the inconsistent spelling of sulfur vs. sulphur (including "hydrogen sulphide"). The authors should use the spelling "sulfur" or "sulfide" as recommended in the publication "So long sulphur" (Nature Chemistry 2009, volume 1, page 333).

6. Font sizes in figures, especially in Extended data and Supplementary Information, are tiny, and the authors should consider enlarging the font.

Reviewer #3:

Remarks to the Author:

Thomas J. M. et al revealed that full-length human CBS in the basal and SAM-bound activated states can form filaments, which is mainly mediated by a conserved regulatory loop 516-525 by determination of high-resolution structures using cryo-EM. These high-resolution structures of full-length human CBS addressed important questions that have not been answered through previous crystal structures. The apo and SAM-bound human CBS full-length protein both can polymerize as a filament adopting two very different morphologies. By combining biophysical and biochemical

methods, they further showed that polymerization stabilizes CBS and increases the cooperativity of allosteric activation by SAM. This study will advance our understanding about CBS enzyme regulation and provide a new route to studying the pathogenic mechanism and for discovering new therapeutics for CBS-associated disorders. In my opinion, this is a well-written manuscript that can be accepted for publication after addressing the following points.

Major Comments:

1. In Fig. 2a, there are two classes showing string filaments, but without the catalytic domain density evident within the filaments. Have the authors observed the filament formed using only the central regulatory domain? What is the interpretation of these data?
2. In discussion. The authors claimed "We theorise though that the His-tag could be acting as a proxy for a as yet unidentified ligand that may regulate CBS oligomerization.". Have any reports suggested that the c-terminal tail of CBS could bind a ligand? Is it possible that the degradation difference between CBSFL and CBSFL-Chis were caused by protein preparation? I noticed the authors prepared these two samples using different purification methods. Is it possible that purified CBSFL has more protease contamination?

Minor issues:

Please correct the typos in sentence 98 of page 2. " i.e., between the RDs of neighbouring s of the filament."

I would recommend to rephrase sentence 208-209 in page 6 to " These mutants showed a basal activity, but their activity could not be stimulated by SAM-binding like WT-type CBSFL-Chis." This would be better, since the authors made mutation in CBSFL-Chis construct for experiments.

Figure issues:

What do labels " A, B " in Fig. 1d mean? Please clarify them in figure legend. I also recommend the authors can color the CBS dimer as in Fig. 1a.

It's better to label α -helix 15 and α -helix 16 and their corresponding displacement in Fig. 2d. For instance: The 8 Å shift in α -helix 15 is not indicated in Fig. 2d. The left arrow in the bottom figure of Fig. 2d which is not pointing to loop 516-525 exactly. It looks like the arrow points to another loop.

Please point out where the S1 and S2 SAM-binding sites are supposed to be in Fig. 2b and Fig. 3a. Otherwise, the reader could be not easy to follow the authors' discussion regarding to S1 and S2 SAM-binding sites.

The figure citations in "Filament formation increases cooperativity of SAM activation and CBS stability" section are incorrect.

For instance,

The authors referred figure citation in sentence 244 should be "Extended Data Fig 7a".

The figure citation in sentence 246 should be "Fig. 4a, Extended Data Fig 7b"

The figure citation in sentence 248 should be "Fig. 4a, Extended Data Fig 7b"

The figure citation in sentence 252 should be "Fig. 4b, Extended Data Fig. 6a"

The figure citation in sentence 258 should be "Fig. 4c"

REVIEWER COMMENTS

Reviewer #1 (Remarks to the Author):

Below I itemize comments made while reading the manuscript that may help the authors in improving readability:

1. In the Abstract. “Human CBS appends to its conserved catalytic domain a regulatory domain”. I understood this sentence the second time I read it, but the first time I read it, it was confusing and sounded like the enzyme CBS had a catalytic activity that appended domains onto itself. Something like “Appended to the catalytic domain is a regulatory domain in human CBS” or more simply, “Human CBS has an additional regulatory domain appended to its catalytic domain”, or something similar.

Response – We thank the referee for their comments and for taking their time to review our submitted manuscript. This sentence has now been edited to read “Appended to the conserved catalytic domain of human CBS is a regulatory domain that modulates ...” to improve readability (page 1).

2. Same sentence, which ends with “... that modulates activity by S-adenosyl-L-methionine (SAM) and promotes oligomerization, however, the molecular basis is unknown”. The molecular basis of what? I assume this means either the molecular basis of regulation, or of oligomerization, or both, but the sentence could be written in a way that is more clear.

Response – Yes, the referee is correct that our wording is unclear. This phrase has now been removed and the full sentence (page 1) reads as “Appended to the conserved catalytic domain of human CBS is a regulatory domain that modulates activity by S-adenosyl-L-methionine (SAM) and promotes oligomerization.”

3. Last sentence of the abstract, “elaborate our understanding” is a strange use of the word elaborate. I would reword.

Response – The word “elaborate” has now been removed and the full sentence (page 1) reads as “Together our findings extend our understanding of CBS enzyme regulation, and open new avenues for investigating the pathogenic mechanism and therapeutic opportunities for CBS-associated disorders.”

4. Introduction, “lost-of-function” should probably be “loss-of-function”.

Response – This error has now been corrected to “Inherited loss-of-function mutations of CBS result in classical homocystinuria (HCU)....” (page 1).

5. Extended Figure 1 A and text on page 2 (lines 58-69), it is hinted by Extended Figure 1A, but not stated anywhere, that there might be two active sites per dimer (Extended Figure 1A shows that the active site of the blue chain is blocked when SAM is not bound, and then when SAM is bound, the active site of shown open, but in the symmetry-related subunit (suggesting two active sites, though it is unclear).

Response – As per Nature Communications formatting, Extended Data Fig. 1 is now Supplementary Fig. 1. We have edited this figure to show that there are two active site per CBS dimer (one active site per protomer). This is shown below:

Additionally, we have added a sentence to the figure legend for clarity over the crystal structure 4PCU: “Note that in the 4PCU crystal structure the Bateman module dimer is tilted relative to the catalytic dimer due to crystal packing and both active sites have increased accessibility.”

6. Extended Figure 2b, the colors are difficult to distinguish (in a printout, but not if read online).

Response – As per Nature Communications formatting, Extended Data Fig. 2b is now Supplementary Fig. 2b. We have thickened line widths to render the different line colours more distinguishable.

7. Line 98 “Here the RD sits atop the active site entrance of the catalytic domain hindering substrate access (Fig. 1c,e)”. Figure 1c doesn’t identify the active site entrance. Perhaps the authors meant Fig. 1f and e?

Response – This error has been corrected to “(Fig. 1e, f)” (page 2).

8. The blue colors in Figure 1e, notably in the text labels, are very difficult to distinguish. They are easily distinguishable in the map as shown here, but not the labels. The blues are more difficult to distinguish in Extended data Fig. 3d-e. This is especially true when in printed form.

Response – Text labels in Figure 1d and e have been reformatted to black for better clarity. As per Nature Communications formatting, Extended Data Fig. 3 is now Supplementary Fig. 7. The model representations in Supplementary Fig. 7d-e are now coloured with more distinguishable blue shades.

9. In Figure 1d, the legend indicates that one subunit is colored as in Figure 1a, however its partner subunit in the dimer is colored white or grey I think. This should be made clear because then the other two dimers shown (2 and 4) are colored as in Figure 1c, although Figure 1c should indicate what the colors mean (the two subunits of the domain-swapped dimer probably?).

Response – The figure legend text for Fig. 1c and 1d has been clarified (page 3).

10. Colors in extended figure 1 (Ed: 3?) should be defined (the two blue colors in b-f). These color choices make it harder to see the difference between the two subunits. The boxed regions in extended data Figure 3b should be connected somehow to the regions shown in d-e to make it clear that the closeup views map to those locations in the zoomed-out view.

Response – We believe the reviewer referred to Extended Data Fig. 3 which is now Supplementary Fig. 7 (as mentioned in point 8 above). We have now changed the color scheme in this figure coloring one of the subunits gray. We hope this increases the contrast to see the difference between the two subunits. Additionally, we have connected the regions shown in d-e to Fig. 3b.

11. Line 182, the reference to Fig. 2d, the next mentions loop 516-525, and residues 422-426 of alpha-helix 15, but these aren't identified in Fig. 2d.

Response – These regions are now labelled in Fig. 2d.

12. line 186, “little clashes” might be better as “fewer clashes”.

Response – The suggested change has been made (page 6).

13. Fig. 3a, they might mention that the dimer interface is being shown, with one dimer colored white and one salmon, hence why two molecules of SAM are shown.

Response – The suggested change has been made in Fig. 3 legend text (page 7).

15. In the paragraph beginning at line 233, references to Extended Data Fig. 6a should be to 7a, and Fig. 3a should be Fig. 4a. Perhaps the extended data figures need to be reordered so that they are called out in order?

Response – Fig. 4a is now cited. Supplementary figure numbering has also been corrected (page 7). Note that Extended Data Fig. 7 is now Supplementary Fig. 14.

16. Are the differences in Hill coefficients statistically significant? The error bars lead these values to overlap. Some other statistical tests could be applied. This could be an important point in the paper, that the filament increases the cooperativity of the enzyme.

Response – The referee is correct in pointing out that statistical tests are needed to determine if the differences in Hill coefficient are statistically significant. We have now applied a Student's T-test to the data for all three CBS constructs that should cooperativity due to SAM binding. Comparison of CBS^{FL} to CBS^{d516-525} gave a p value of 0.2218 while comparison of CBS^{FLC-His} to CBS^{d516-525} gave a p value of 0.3382. These show that the Hill values are not significantly different from each other. Therefore, we have rewritten the section (page 7) on possible cooperativity, added these P values, and toned down the discussion (page 11). Further studies are likely needed to determine if there is indeed an effect on SAM cooperativity of CBS.

17. The paragraph beginning on line 249, references to extended data Fig. 5a should be to 6a. also, it would be really difficult to read this figure in printed form due to the same size of the text fonts.

Response – Supplementary figure numbering has been corrected in the text (Note that Extended Data Fig. 6 is now split into Supplementary Figs. 12 and 13). We have also increased font size and magnified figure panels to improve readability.

18. In the results and discussion, the authors argue that the second observed binding of SAM at 600 nM is actually not a binding event, but heat dissipation from the conformational change of the enzyme in the filament. Why should it occur at this exact concentration of SAM every time? Is that a function of how the ITC was done (the authors argue that it takes time for the conformational change to occur after SAM binding)? If titrations were done that did not reach 600 nM SAM, would this heat dissipation still occur as expected? If the titration were slowed, would the heat dissipation occur at a different concentration of SAM?

Response – We apologise for the lack of clarity, but we mean that both events likely correspond to a mix of conformational changes and SAM binding. We have therefore changed the sentence on page 6 to “We therefore reasoned that the two apparent binding events could be attributed to a combination of both the binding of one SAM to the S2 site and the resulting conformational

rearrangement into the activated state.” In terms of the ITC of CBS^{FL-CHis}, CBS^{FL}, and CBS^{Δ516-525} we cannot attribute individual events to only exclusively conformational changes or ligand binding. Also, we would not have the confidence to state that the lower binding event is ligand binding based off the results of CBS^{Δ516-525} and have not done so. To make this clear we have also added more details to the ITC section (page 6) such as highlighting that CBS^{Δ516-525} shows two events though this construct only forms dimers and as such wouldn't form a putative cryptic SAM binding site at a higher order oligomeric interface.

We also have not made any statements on the timeframe for the SAM to induce the CBS conformational change and such studies are beyond the scope of our present study. We do not know why the events occur at the same concentration of SAM every time. The setup of the ITC was based off the work of Pey et al 2013 (<https://doi.org/10.1042/BJ20120731>) who initially showed two binding events however their interpretation was based on the proposed tetrameric model of CBS and only the basal state (no SAM) crystal structures of CBS^{CD} and CBS^{Δ516-525} were available at the time. Of course, more complicated changes to the ITC setup could possibly show more details into the SAM response of CBS but again is beyond the scope of our study. As ITC is an equilibrium-based technique titrating amounts of SAM that would not reach 600 nM would likely only show the first event. Additionally slowing down the titrations by increasing the spacing would likely not change the resulting heat dissipation. However, these are just assumptions, and we will take them into consideration for future studies.

19. The authors show that mutation of the first SAM binding site eliminates both binding events, and that the construct with only the RD (and no regulatory loop) has only 1 and argue that these results are evidence that the second so-called binding event is actually a conformational change. Could it not also be that the second binding site is not in the residues in the RD construct, but does overlap the first SAM binding site such that the mutations affect its binding as well?

Response – Again we apologise for the lack of clarity on these points. There are no statements in our manuscript that says that the second so-called binding event is exclusively a conformational change, but we understand why the reviewer has come to these conclusions due to our unclear writing. We meant that both events likely correspond to a mix of conformational changes and SAM binding for CBS^{FL-CHis}, CBS^{FL}, and CBS^{Δ516-525}. The RD construct in comparison to the CBS^{Δ516-525} removes any complicating factors due to the conformational change (in relation to the catalytic domain) and shows only one binding event that aligns with our previous crystal structure of this construct (PDB 4UUU). Importantly we have not attempted to attribute the two binding events of CBS^{FL-CHis}, CBS^{FL}, and CBS^{Δ516-525} to separate binding events and conformational changes considering the ITC results of CBS^{RDΔ516-525}. We hope that our edits described in the previous response (point 18) clarifies this.

However in regards to the reviewer's second point - Yes it could be possible, as the reviewer alluded, that the RD-only construct does not have an additional site that would be formed by the oligomerisation of CBS as hypothesised in Pey et al 2013 and Pey et al, 2016 <https://doi.org/10.1002/1873-3468.12488>. It is possible that the mutation of the S2 site could affect any other cryptic putative site. However, based off our structural information of CBS^{FL-CHis} in the presence of saturating SAM (5 mM) where we expect to see both high affinity stabilising sites and low affinity activating sites occupied, we only see SAM density at the previously determined S2 site (previously suggested by Pey et al, 2013 and Pey et al, 2016 to be the second low affinity binding site that activates CBS). Hence this is the reason we have attempted to interpret the ITC results considering our structural results. We do recognise that the SAM bound map is of lower resolution than our basal state maps and so are more difficult to interpret. It is possible that a high affinity site is at the interface between the regulatory domains that is flexible in nature and difficult to see using cryo-EM. We have adjusted the writing on the results (page 6), and discussion (page 10) sections considering that mutation of the S2 site could affect a putative cryptic site.

20. The ITC results could be presented and discussed more to show for one thing how strong the data are for the two binding events and how to interpret them. Also, why are the enthalpies and entropies so different for CBSFL and CBS-FL-Chis with respect to the first SAM binding site (Ext. Fig. 6d)?

What are the measures of quality of fitting of the ITC data? I assume the red line in the delta H vs. Molar Ratio is the fit, while the points are the experimental data? This should be explained in the legend.

*Response – We have now added the residuals of fit for all ITC results to show the quality of fit for the ITC data, in Supplementary Figs. 12 & 13. These show good fit to their respective models. It is unknown for sure why the enthalpies and entropies are so different for the first binding event when comparing the full-length constructs of CBS. We assume it may be due to the C-terminal His-tag present for the CBS^{FL-CHis} construct. This His-tag is present at the oligomeric interface and may face the neighbouring dimer's His-tag. As we see that the C-terminal His-tag influences both the length of the filaments and stability the differences between it is possible that these effects are linked to the differences see in ITC for the first binding event. We have now adding the following sentence to the ITC section (Page 6) to highlight this difference and hypothesise on why this effect is seen: “**Oddly the first binding event of CBS^{FL-CHis} is endothermic (positive enthalpy) and increases order (negative entropy), in contrast to the constructs without a C-terminal His-tag which have a first binding event that is more exothermic (negative enthalpy) and increases disorder (positive entropy) (Supplementary Figs. 12b and 13). Though we do not know the exact reason for this phenomenon, this difference is likely due to the longer oligomers and stabilising effect of the C-terminal His-tag (Supplementary Figs. 2 and 12a).***

We have stated in Fig. 3 legend (Page 7) that the red line is the fit of the ITC data to the two-site binding model.

20. Fig. 5a, is another figure with very small fonts making it difficult to read, especially when in print without the option to zoom.

Response – We have increased all font size to improve readability of the now Fig. 6 panels.

Reviewer #2 (Remarks to the Author):

Due to limited expertise in structural biology, the reviewer cannot critically assess the methods, results, and bioinformatic interpretation of CBS quinary structure. Here are several comments to consider:

1. Relevance of in vitro filament formation for in vivo biology. The study used purified CBS protomers to demonstrate the formation of CBS fibrils under non-physiological conditions. The critical question is whether CBS fibrils also form in vivo in human cells. The authors provided only one indirect supporting indication for human CBS filamentation in the cellular environment, i.e., the formation of punctate foci of the yeast CBS ortholog (encoded by the *CYS4* gene) during the stationary phase. It is unlikely that the present study will be expanded experimentally to prove the filamentation of human CBS in cellular systems; however, the authors should discuss future directions to confirm the filamentation of wild type human CBS in vivo.

Response – We thank the reviewer for motivating us to consider CBS filamentation/polymerization in human cells. During the revision process, we reached out for collaboration to the group of Dr Sandra Martha Gomes Dias (LN Bio, Campinas) who have carried out fluorescence microscopy on human cells transfected with labelled CBS. The results, presented in the new Fig. 5 and accompanying text on page 8, clearly showed an aggregated pattern of fluorescence signal in the case of full-length CBS, in agreement with CBS polymerization. The fluorescence signal becomes more punctuate when methionine is depleted from the medium but can be reversed when SAM is supplemented. Importantly, the aggregated signal is lost completely in the case of CBS^{Δ516-525}. Altogether, this provides the first demonstration of CBS polymerization in cellulo and reinforces the essentiality of loop 516-525 in this. We have also provided a paragraph to discuss these implications on page 11.

2. Size of oligomers and length of filaments. The study shows that the filaments are assembled from dimers. The estimated size of the dimer is about 120 kDa. Can the authors comment on the differences between the predicted molecular weight of the dimer and the apparent size on Superose

gel filtration (major peak size at 15 ml, i.e., between 158 and 670 kDa), clear native gel (major signal between 242 and 480 kDa) and BN gel (major signal between 146 and 242 kDa)? In addition, I could not find information on the lengths of fibrils and the underlying number of subunits; the authors may consider adding this information.

Response – The differences in MW estimation between the analytical gel filtration, blue native (BN) PAGE, and clear native PAGE are likely due to each technique using different matrices and methods to separate oligomers. The discrepancy in the molecular weight for the dimer species from approximately 120 kDa is because these techniques are separating protein species in terms of not only their native (non-denatured) size but, in relation to native PAGE, also surface charge. Inherent errors in measurements also partially contribute. Differences between the blue and clear native PAGE are also likely due to the presence of the G-250 Coomassie dye added to the cathode buffer during running of the gel. The dye can bind to proteins (<https://doi.org/10.1007/s00216-008-1996-x>) and likely partially breaks the CBS oligomers apart explaining the smaller molecular weight but clearer bands in BN PAGE.

We have now added, to Supplementary Fig. 2, the calculated molecular weight and potential number of monomeric CBS subunits for each of the peaks for the various CBS constructs. These calculations are based off the molecular weight standards that were ran on the size exclusion column. These values give some indication of the size of the oligomers of CBS and the number of monomers in each CBS species. These show that the full-length without a His-tag, CBS^{FL}, forms oligomers (Peak 1) consisting of up to 290 monomers (145 dimers) while CBS^{FL-His} forms even larger oligomers (Peak 2) of possibly up to 600 monomers (300 dimers). The additional peak (Peak 1) of CBS^{FL-His} is at the void volume of this column and tentatively even larger oligomers are present however this measurement is complicated by the likely presence of aggregates and should be treated with caution. We also observe a lower molecular peak (Peak 2) for CBS^{FL} that likely corresponds to the degraded tetramer that we have observed in cryo-EM micrographs and have determined the cryo-EM structure of. All other calculated molecular weights are within an acceptable error and align with expected molecular weight of each CBS construct.

To determine the length of the filaments we attempted negative stain electron microscopy (EM), initially on the CBS^{FL} and CBS^{Δ516–525} constructs. This technique uses low concentrations of protein by applying it to an EM grid covered in a thin layer of carbon and then staining with the heavy metal stain uranyl acetate to give high contrast but low-resolution images on an electron microscope. These low-resolution and high contrast images would be ideal for measuring filament length. Unfortunately, we found the CBS^{FL} construct to be quite heterogenous, and it was difficult to interpret any higher order oligomers. It is likely that the low concentration of protein (in the nanomolar range, 50 nM) and the uranyl acetate dye contributed to the disassembly of any large oligomers, like the effect of the Coomassie dye in BN PAGE. Additionally, the flexible nature of the CBS oligomers makes it difficult to discern individual dimer units and measure their length. Further optimisation is clearly needed. In contrast, CBS^{Δ516–525} presented better behaviour and showed individual particles of the expected dimensions for this dimeric construct (approximately 9 nm long based off crystal structure PDB 4COO). Below we show representative micrographs imaged on a 120 kV Hitachi HT7800 electron microscope of both constructs at the same magnification.

Some individual dimers (red arrow), large clumps of protein larger than 9nM (yellow arrow) Clear individual dimers (red arrow), larger round clumps are likely aggregates or artifacts from the stain

Individual CBS dimers have a diameter of approximately 9nM based off CBS^{Δ516-525} crystal structure (PDB 4COO).
Scale bar in micrographs= 50 nm

Due to these issues, we have attempted to measure the oligomers from our cryo-EM micrographs, which are of much lower contrast. As this is not an extensive analysis, we present it only in the below:

Representative Cryo-EM Micrographs of CBS^{FL} and CBS^{Δ516-525}

White scale bar = 50 nm

Contrast has been increased of both micrographs

From these representative cryo-EM micrographs the flexible nature of the CBS oligomers is clear, along with some clumping, but individual dimeric units are somewhat clearer than what is observed in negative stain EM. Each individual dimer can be seen as a “U” shape dependent on its orientation in the ice. Therefore, the CBS oligomers present themselves as flexible coils like that of the wire of a wired telephone. As each of these “U” shapes is approximately 9 nm in length we can approximate the filaments’ total lengths (the yellow lines). It is however sometimes difficult to discern where a filament ends when it is close to another. For the CBS^{FL-CHis} construct, the oligomers are approximately in the range of 45 to 300 nm, 5 to 23 dimer subunits, and approximately 600 to 2,760 kDa. There are however longer filaments. For CBS^{FL} the oligomers are smaller as expected from the solution analysis of this construct in comparison to the CBS^{FL-CHis} construct. They vary from approximately 18 to 180nm, 2 to 20 dimer subunits, and approximately 240 to 2,400 kDa. Again, there are longer filaments, but they are difficult to discern, measure, and count individual subunits.

This contrasts with our analytical gel filtration analysis that suggests longer (up to approximately 1,323 nm) and larger oligomers (up to approximately 3,700 kDa) for both constructs. The CBS filaments are clearly very flexible and prone to disassembly and the differences may be due to how cryo-EM grids are prepared: they are glow discharged, which may cause disassembly of larger oligomers and solutions of protein on the grid are blotted which may remove some of the larger oligomers and aggregates. However, we believe our electron microscopy results are consistent in terms of oligomer formation of full-length CBS at the regulatory domain and that the C-terminal His-tag has an influence on oligomerisation. We hope that these answer the questions of the reviewer, and we inform the reviewer that we will be looking more in depth into filament formation in the future of homocystinuria related mutants.

3. Pathogenicity of the p.D444N variant. The authors show in Figure 3 the model of the SAM S2 binding site also containing a well-studied pathogenic mutation p.D444N. The pathogenicity of this variant is intriguing as it is constitutionally active to overactive and cannot be further activated by physiologically relevant SAM concentrations (e.g. reference 34, reference Kluijtmans et al. Defective cystathionine beta-synthase regulation by S-adenosylmethionine in a partially pyridoxine responsive homocystinuria patient. Clin Invest. 1996 Jul 15;98(2):285-9 and reference Evande et al. Alleviation of intrasteric inhibition by the pathogenic activation domain mutation, D444N, in human cystathionine beta-synthase. Biochemistry 2002 Oct 1;41(39):11832-7). Using their model, can the authors shed light on the possible pathogenic mechanism(s) of the p.D444N mutation?

Response – Yes, it is very interesting to hypothesise on the effects of the many HCU mutations of CBS through the lens of our new structural information. p.D444N is one of a few HCU mutations found in the regulatory domain and appears to be involved in SAM binding. The reviewer points to two papers that we have referenced, Kluijtmans et al and Evande et al, that have characterised p.D444N showing this mutant has a higher basal activity and a reduced ability to be activated by SAM. We believe that the pathogenic mechanism is possibly multi-faceted, however in terms of our new model we refer to the previous determined crystal structure by Ereño-Orbea et al 2013. The mutation from aspartate to asparagine may destabilise the interactions between the regulatory and catalytic domains in the basal state as has been shown by the crystal structure of this mutant in the context of the CBS construct without the loop 516-525: <https://www.rcsb.org/structure/4l28>. This crystal structure shows that p.D444N results in shift of the strand 8 and helix 15 of the regulatory domain that form part of this newly discovered Bateman-Bateman interface.

Below we show a structural alignment of the 4L28 crystal structure (orange) with our cryo-EM basal state model (cyan):

The p.D444N mutation may affect the oligomerisation of CBS due to these structural changes resulting in decreased stability. As such we are very interested in characterising and determining the structures of multiple HCU associated mutations, p.D444N being one of them, along with understanding any link to filament formation. However, we believe it is beyond the scope of our submitted manuscript to go into the details of the possible effects of individual disease associated mutations. Future work may give clearer insights into D444N and other HCU mutants as stated in our last paragraph of the Discussion.

4. Suppression mutations. The authors state: “It is intriguing that genetic suppression in a yeast model of the disease has also been reported, where deletion²⁹ or missense mutations on the regulatory domain²⁹ can overcome the deleterious effects of the most common HCU mutations.” The reference for suppression by missense mutation seems incorrect and should be replaced by reference 28. Moreover, the statement is incorrect as reference 28 studied functional correction of only two mutants (p.I278T and p.V168M). Please, revise accordingly.

Response – We have now corrected the error of using the incorrect reference in regards to the suppressor mutations of the regulatory domain (page 11). We have also revised our statement of “the most common HCU mutations” to “some HCU mutations”, on page 12.

5. Sulfur spelling. The authors use the inconsistent spelling of sulfur vs. sulphur (including “hydrogen sulphide”). The authors should use the spelling “sulfur” or “sulfide” as recommended in the publication “So long sulphur” (Nature Chemistry 2009, volume 1, page 333).

Response – The spelling of sulfur and sulfide is now used throughout the manuscript.

6. Font sizes in figures, especially in Extended data and Supplementary Information, are tiny, and the authors should consider enlarging the font.

Response – Where applicable, we have now increased font size for supplementary figures.

Reviewer #3 (Remarks to the Author):

Major Comments:

1. In Fig. 2a, there are two classes showing string filaments, but without the catalytic domain density evident within the filaments. Have the authors observed the filament formed using only the central regulatory domain? What is the interpretation of these data?

Response – In Fig. 2a we have shown 2D classes from preliminary processing of this CBS construct using a Glacios (200 keV) microscope. We interpret these classes as perpendicular views of these filaments. Here one view shows the central regulatory domain filament in the centre with flexible catalytic domains. As these filaments have a twist of approximately -180 degrees the perpendicular view is of the catalytic domains which obscure the central regulatory domain underneath.

However, it is an interesting aspect to consider whether the RD alone can form filaments, as CBS is known to be cleaved in a region at the linker region (Kery et al, 1998:

<https://doi.org/10.1006/abbi.1998.0723>). We have not observed any particles nor 2D classes that would correspond to just the SAM bound regulatory domain alone for the CBS^{FLC-His} construct. It could be possible that they are too small to pick or classify. We have also attempted many times to produce recombinant regulatory domain with the oligomerising loop however all constructs have been insoluble hindering any further studies.

2. In discussion. The authors claimed “We theorise though that the His-tag could be acting as a proxy for a as yet unidentified ligand that may regulate CBS oligomerization.”. Have any reports suggested that the c-terminal tail of CBS could bind a ligand? Is it possible that the degradation difference between CBS^{FL} and CBS^{FL-CHis} were caused by protein preparation? I noticed the authors prepared these two samples using different purification methods. Is it possible that purified CBSFL has more protease contamination?

Response – We thank the referee for raising these points. As far as we are aware there are no reports of a ligand binding to the C-terminal tail of CBS. However, there are reports of this region being involved in the SAM responsiveness of CBS. Oliveriusova et al 2002 reported that deletion of the final 8 amino acids at the C-terminus of CBS results in a SAM non-responsive enzyme:

<https://doi.org/10.1074/jbc.M207087200>. Based off our cryo-EM structures of the full-length CBS (and AlphaFold predicted structures) it is difficult to rationalise the structural effect of these deletions. This region could be binding either an unknown ligand or possibly SAM, but we are limited by the resolution and quality of our EM maps of CBS in the presence of SAM.

It is possible that the degradation difference between the CBS^{FL} and CBS^{FL-CHis} is due to the protein preparation. It is also possible that the C-terminal His-tag enriches more full-length protein as the major protease site is known to be the flexible linker region between the CD and RD as reported by Kery et al, 1998: <https://doi.org/10.1006/abbi.1998.0723>. Additionally, the use of a C-terminal His-tag construct has been previously reported to increase the yield of CBS by the group of the late Prof Jan Kraus: <https://doi.org/10.1016/j.pep.2012.01.019>.

Minor issues:

Please correct the typos in sentence 98 of page 2. “ i.e., between the RDs of neighbouring s of the filament.”.

Response – This is now corrected.

I would recommend to rephrase sentence 208-209 in page 6 to “ These mutants showed a basal activity, but their activity could not be stimulated by SAM-binding like WT-type CBSFL-Chis.” This would be better, since the authors made mutation in CBSFL-Chis construct for experiments.

Response – This sentence has been rephrased based on suggestion.

Figure issues:

What do labels “A, B ” in Fig. 1d mean? Please clarify them in figure legend. I also recommend the authors can color the CBS dimer as in Fig. 1a.

Response – We have now explained A,B as well as its colouring in the legend text of Fig. 1 (page 3).

It's better to label α -helix 15 and α -helix 16 and their corresponding displacement in Fig. 2d. For instance: The 8 Å shift in α -helix 15 is not indicated in Fig. 2d. The left arrow in the bottom figure of Fig. 2d which is not pointing to loop 516-525 exactly. It looks like the arrow points to another loop.

Response – We have now edited this figure with the additional labels as requested, along with clarifying the position of the arrows.

Please point out where the S1 and S2 SAM-binding sites are supposed to be in Fig. 2b and Fig. 3a. Otherwise, the reader could be not easy to follow the authors' discussion regarding to S1 and S2 SAM-binding sites.

Response – As per the referee's recommendation to help readers in following our discussion on both the S2 and S1 sites, the S1 and S2 sites have now also labelled in Fig. 2c as “Occupied S2” and “Empty S1”. In Fig. 3a each SAM molecule is now labelled as “SAM (S2)”. As the S1 site is behind the S2 site on the other face of the CBS regulatory domain we believe that it would be difficult to indicate S1 on Fig. 2b and Fig. 3a. Instead, we have labelled the putative S1 site on Supplementary Fig. 1a, hoping this will suffice in aiding readers.

The figure citations in “Filament formation increases cooperativity of SAM activation and CBS stability” section are incorrect.

For instance,

The authors referred figure citation in sentence 244 should be “Extended Data Fig 7a”.

The figure citation in sentence 246 should be “Fig. 4a, Extended Data Fig 7b”

The figure citation in sentence 248 should be “Fig. 4a, Extended Data Fig 7b”

The figure citation in sentence 252 should be “Fig. 4b, Extended Data Fig. 6a”

The figure citation in sentence 258 should be “Fig. 4c”

Response – These errors have now been corrected. The referee should note that the manuscript has been reformatted as per Nature Communications, and therefore all extended data figures are now incorporated as supplementary figures (and may be renumbered). For example, Extended Data Fig. 6 is now Supplementary Figs. 12 & 13, and Extended Data Fig. 7 is now Supplementary Fig. 14.

Reviewers' Comments:

Reviewer #1:

Remarks to the Author:

The authors have addressed all of my points in my review.

Reviewer #2:

Remarks to the Author:

I congratulate the authors on the revision. I appreciate their efforts on demonstrating the formation of CBS fibrills in human cells. All my comments were addressed adequately, there are no additional issues to be raised.

Reviewer #3:

Remarks to the Author:

By employ Cryo-EM to determine the filamentous structures of full-length human CBS in the basal and SAM-bound activated states. Thomas J. M. et al revealed that a conserved regulatory loop 516-525 in full-length CBS is the major driving force for filamentation of human CBS, the basal and SAM-bound filaments adopt two different morphologies. Combining biophysical and biochemical methods, they further showed that polymerization increases the thermal stability of the basal state conformation of CBS and the cooperativity of allosteric activation by SAM. This study eliminating some conflicting reports will advance our understanding about the mechanism of CBS enzyme activity regulation and provide a new route for discovering new therapeutics for CBS-associated disorders. In this well-written revised version, their cellular data also demonstrated that CBS filamentation is responsive to nutrient changes. In my opinion, the data was interpreted properly in this revised manuscript. I highly recommend it for publication in Nature Communications after following minor issues are addressed.

Minor comments:

1. Line 118 – 120 in page 3, the authors mentioned the shift of both Ala421 and Pro422 in their full-length human CBS structures, compare with previous crystal structures of CBS Δ 516–525 . Would the author label the fragment containing Ala421 and Pro422 in their structure in Fig. 7f to show a clear shift?
2. I'm just curious about the degradation of full-length CBS FL in the context of its biological functions. Do the authors have any insight on this? Does the degradation of full-length CBS FL have any relations with CBS-related disease development? Could the authors elaborate on these questions a little bit on the section of " CBS degrades into tetramers and dimers"?
3. Line 207 – 208 in page 6. The figure citation is not correct. The authors should cite Figs. 2d, 3a for this sentence.
4. It's surprising to me, only one single mutation of either F443A and D538A could completely abolish the SAM binding to S2 site. Just comment on it, no any further action is needed.
5. Correct the typo in line 432. "where deletion of missense mutations ..."

REVIEWER COMMENTS

Reviewer #1 (Remarks to the Author):

The authors have addressed all of my points in my review.

Reviewer #2 (Remarks to the Author):

I congratulate the authors on the revision. I appreciate their efforts on demonstrating the formation of CBS fibrils in human cells. All my comments were addressed adequately, there are no additional issues to be raised.

Reviewer #3 (Remarks to the Author):

By employ Cryo-EM to determine the filamentous structures of full-length human CBS in the basal and SAM-bound activated states. Thomas J. M. et al revealed that a conserved regulatory loop 516-525 in full-length CBS is the major driving force for filamentation of human CBS, the basal and SAM-bound filaments adopt two different morphologies. Combining biophysical and biochemical methods, they further showed that polymerization increases the thermal stability of the basal state conformation of CBS and the cooperativity of allosteric activation by SAM. This study eliminating some conflicting reports will advance our understanding about the mechanism of CBS enzyme activity regulation and provide a new route for discovering new therapeutics for CBS-associated disorders. In this well-written revised version, their cellular data also demonstrated that CBS filamentation is responsive to nutrient changes. In my opinion, the data was interpreted properly in this revised manuscript. I highly recommend it for publication in Nature Communications after following minor issues are addressed.

Minor comments:

1. Line 118 – 120 in page 3, the authors mentioned the shift of both Ala421 and Pro422 in their full-length human CBS structures, compare with previous crystal structures of CBS Δ 516–525 . Would the author label the fragment containing Ala421 and Pro422 in their structure in Fig. 7f to show a clear shift?

Response – We have added arrows in Supplementary Fig. 8f (was previously Supplementary Fig. 7f) to indicate the shift of Ala421 and Pro422.

2. I'm just curious about the degradation of full-length CBS FL in the context of its biological functions. Do the authors have any insight on this? Does the degradation of full-length CBS FL have any relations with CBS-related disease development? Could the authors elaborate on these questions a little bit on the section of “ CBS degrades into tetramers and dimers”?

Response – We have little insight into the biological function of the degradation apart from that it has been observed by many other researchers who study CBS. It could very well be related to classic homocystinuria, as many disease mutants cause misfolding and not only cause aggregation but degradation as well. It could also be a consequence of recombinant expression in bacteria. As such we have added the following sentence at the end of the relevant section (page 3): “It is unknown if this degradation is due to recombinant expression or is related to a biological function of CBS. Further studies are needed to determine its significance.”

3. Line 207 – 208 in page 6. The figure citation is not correct. The authors should cite Figs. 2d, 3a for this sentence.

Response – Thank you for spotting this error. It is now corrected to Figs. 2d, 3a (page 4).

4. It's surprising to me, only one single mutation of either F443A and D538A could completely abolish the SAM binding to S2 site. Just comment on it, no any further action is needed.

Response – We have previously studied these and other residues that interact with SAM in our crystallographic work on CBS (<https://doi.org/10.1074/jbc.M114.610782>). At the time we did not have an activity assay, nor did we do ITC; however we analysed these residues by a combination of alanine mutagenesis followed by limited proteolysis and thermal shift. Out of the six mutants analysed both F443A and D538A showed the greatest effects in terms of a reduced ability to be thermally stabilised, and showed no increase in susceptibility to thermolysin in the presence of SAM. We interpreted the thermal shift results as a reduced ability to bind SAM and the proteolysis suggested that these mutants could not change conformation into the SAM-bound activated state. Our present manuscript follows up on these results where the activity analysis of these mutants agrees with the previous findings. The ITC experiment is not surprising where these two mutants show no detectable binding of SAM, but it is important to note that the highest concentration of SAM used was 500 μ M. This is enough to saturate the wild-type enzyme and activate it. It is possible that these mutants could be activated and show binding with far higher amounts of SAM. Indeed, our thermal shift experiments from 2014 (above reference link) suggest that SAM can stabilise both these mutant but only at concentrations higher than 700 μ M. We will certainly investigate higher concentrations of SAM in any future ITC and activity assays on these and other mutants.

5. Correct the typo in line 432. “where deletion of missense mutations ...”

Response – Thank you for spotting this typo. It is now corrected.